# Analyzing Memorization in Large Language Models through the Lens of Model Attribution

**Tarun Ram Menta**[*]
Adobe MDSR
menta.tarun@gmail.com

**Susmit Agrawal**[*]
IIT Hyderabad
susmit.agrawal@iith.ac.in

**Chirag Agarwal**
University of Virginia
chiragagarwal@virginia.edu

## Abstract

Large Language Models (LLMs) are prevalent in modern applications but often memorize training data, leading to privacy breaches and copyright issues. Existing research has mainly focused on post-hoc analyses—such as extracting memorized content or developing memorization metrics—without exploring the underlying architectural factors contributing to memorization. In this work, we investigate memorization from an architectural lens by analyzing how attention modules at different layers impact its memorization and generalization performance. Using attribution techniques, we systematically intervene in the LLM's architecture by bypassing attention modules at specific blocks while keeping other components like layer normalization and MLP transformations intact. We provide theorems analyzing our intervention mechanism from a mathematical view, bounding the difference in layer outputs with and without our attributions. Our theoretical and empirical analyses reveal that attention modules in deeper transformer blocks are primarily responsible for memorization, whereas earlier blocks are crucial for the model's generalization and reasoning capabilities. We validate our findings through comprehensive experiments on different LLM families (Pythia and GPT-Neo) and five benchmark datasets. Our insights offer a practical approach to mitigate memorization in LLMs while preserving their performance, contributing to safer and more ethical deployment in real-world applications. The code and data for our work are publicly available.

## 1 Introduction

Large Language Models (LLMs) have become ubiquitous in modern applications, powering everything from conversational agents to advanced data analysis tools. However, these models often memorize parts of their training data, causing serious concerns such as privacy and copyright infringements (Carlini et al., 2020; Huang et al., 2022; Lee et al., 2023; Ishihara and Takahashi, 2024), hindering its employment in high-stakes applications.

Prior works approach memorization from a post-hoc perspective, where they predominantly focus on methods to extract memorized samples from trained LLMs (Carlini et al., 2020; Huang et al., 2022) or propose metrics to quantify memorization (Carlini et al., 2022; Ishihara and Takahashi, 2024). While some works also explore machine unlearning techniques to remove specific memorized content after training (Huang et al., 2024), they do not delve into the fundamental model mechanisms that contribute to memorization. There has been little to no exploration of understanding memorization from a fundamental architectural viewpoint. In particular, systematically analyzing which layers and modules within transformer architectures contribute to memorization. We argue that identifying these components is essential for developing strategies to mitigate memorization without compromising the model's generalization capabilities.

**Present work.** In this work, our goal is to understand the trade-off between reasoning and memorization within LLMs by leveraging model attribution techniques. In particular, we perform systematic architectural interventions by bypassing (or "short-circuiting") the attention modules at targeted blocks of the transformer while preserving other modules such as layer normalization and MLP operations, allowing us to isolate the impact of specific attention modules on both memorization and generalization. We support our experimental findings with theoretical analysis of the impact of bypassing attention modules at different blocks. Our theorems introduce bounds on the difference in model outputs with standard attention and short-circuited attention. Our theoretical insights and

---

[*]Equal Contribution. Work done by the first two authors as Visiting Researchers in **Aikyam Lab**, University of Virginia.

empirical results indicate that short-circuiting attention modules in earlier blocks leads to significant differences in the output vectors compared to the original model and these differences disrupt the model's internal representations, directly hindering its performance and affecting generalization capabilities. In contrast, bypassing attention modules in deeper blocks results in smaller differences in the output vectors, where these changes are sufficient to prevent the model from generating memorized content verbatim and do not significantly impair its ability to generalize from the data.

Our empirical analysis across two different model families—Pythia and GPT-Neo—and four benchmark datasets align with the theoretical findings and demonstrate that intervening in the attention modules of deeper layers mitigates memorization and preserves their generalization capabilities. Our contributions can be summarized as follows:

- We introduce a method to bypass attention modules in transformers while maintaining other transformations like layer normalization and MLP operations. This targeted intervention allows for precise analysis of different components within the model.
- Our theoretical framework explains how differences in output representations caused by bypassing attention modules at various depths impact memorization and generalization and provide evidence that memorization in LLMs primarily stems from the attention modules in deeper transformer layers.
- We offer a practical approach to enhance the ethical deployment of LLMs by demonstrating that it is possible to mitigate memorization while preserving generalization capabilities by bypassing attention modules in deeper blocks.

## 2  Related Work

**Memorization and Extraction of Training Data.** Works on image classifiers argue that a certain degree of memorization is inevitable during generalization (Arpit et al., 2017; Feldman, 2020). This phenomenon is even more pronounced in LLMs with huge parameter counts. LLMs have been shown to memorize and reproduce verbatim samples from their training corpus (Carlini et al., 2020; Nasr et al., 2023). Various works have studied this phenomenon, from the angle of model scale (Ishihara and Takahashi, 2024; Biderman et al., 2024), and data deduplication (Lee et al., 2021). Recent work (Prashanth et al., 2024) has

studied memorization in LLMs with a deeper emphasis on semantics and other factors contributing to memorization. However, very few works explore the architectural aspect of memorization in large transformers. (Stoehr et al., 2024) shows that while memorization cannot be localized to any part of the transformer, predictable gradient patterns are observed. In our work, we show via an attention-bypassing technique that certain blocks of the transformer can be identified as pertinent for memorization, while not boosting the generalization capabilities of the model.

**Interpretability and Pruning in LLMs.** Many attempts have been made to study the inner workings of transformers (Elhage et al., 2021), attribute certain behaviors to individual components (Kissane et al., 2024), and understand the mechanisms of reasoning (Ye et al., 2024) and knowledge storage (Allen-Zhu and Li, 2023a,b, 2024) in LLMs. Pruning of LLMs has also emerged as a key area of interest, with multiple works (Ma et al., 2023; Sun et al., 2023; Siddiqui et al., 2024) showing that LLMs can be pruned in both width and depth without significant performance loss. To our knowledge, our work is the first that attempts to extend such an investigation to memorization in LLMs. We attempt to attribute memorization to individual model components and study if generalization and memorization can be disentangled.

## 3  Methodology

Our goal is to study the phenomenon of memorization in LLMs from the perspective of model attribution. Previous literature (Kim et al., 2024; Ma et al., 2023; Sun et al., 2023) has shown the efficacy of pruning both width and depth of transformers while maintaining downstream performance. This motivates us to study the effect of similar interventions on the memorization characteristics of LLMs. In particular, we wish to isolate the contribution of **individual** model components to memorization. The attention mechanism is fundamental to transformer models, enabling them to understand context and manage long-range dependencies (Elhage et al., 2021). By dynamically weighing the significance of different words relative to each other, transformers capture nuanced meanings and relationships within data. For these reasons, we posit that the major contribution of memorization in LLMs stems from attention, and focus our analysis around it. We develop a mechanism to "short-circuit" the contribution of the attention

mechanism in various blocks of a trained model during, and use it to conduct our experiments.

## 3.1 Preliminaries

Here, we formally define memorization and provide a primer to attention mechanism.

**Definition of Memorization.** Memorization can be defined in various ways within language modeling. Memorization does play a crucial role in language models since the training objective focuses on maximizing the overall likelihood of the training dataset. For example, memorization of well known facts is a desirable characteristic of LLMs. On the other hand, LLMs have been shown (Carlini et al., 2020) to spuriously memorize personal or private data like names, phone numbers, and email addresses; IRC conversations; code; and 128-bit UUIDs. Given a language model $f_\theta$, trained on a set of documents $\mathcal{D}$, we follow the definition of *extractable memorization* set out in previous works (Carlini et al., 2022; Prashanth et al., 2024). A string having a prefix $p$ of tokenized length $l_p$ and a suffix $s$ of tokenized length $l_s$ such that the concatenated sequence $(p||s) \in \mathcal{D}$ is a substring of some document in the training corpus is said to be extractable if the suffix can be recovered by prompting the LLM with the prefix under greedy sampling (GS), *i.e.,* $\text{GS}(f_\theta, p) = s$. In this work, we only explore greedy sampling. Extension to other more directed extraction attacks (Nasr et al., 2023; Yu et al., 2023) is left as future work.

**Primer on Attention Mechanism.** In this section, we setup the notation for the remainder of this work, and provide a brief refresher on the architecture and inner working of a transformer model. We denote a decoder-only transformer LLM as $f_\theta$. $f_\theta$ consists of $L$ transformer layers/blocks in a sequential manner, where the computation of the $l^\text{th}$ block is as follows

$$\mathbf{X}' = \mathbf{X} + \text{MultiHeadSelfAttention}_l(\mathbf{X}) \quad (1)$$

$$\mathbf{X}'' = \mathbf{X}' + \text{FFN}_l(\mathbf{X}') \quad (2)$$

The FFN layer is a simple two-layer MLP with a non-linearity. The main focus of our work is on the multi-head self-attention mechanism. The computation of a single self-attention operation is as follows:

$$\mathbf{Q} = \mathbf{X}\mathbf{W}_\mathbf{Q}; \quad \mathbf{K} = \mathbf{X}\mathbf{W}_\mathbf{K}; \quad \mathbf{V} = \mathbf{X}\mathbf{W}_\mathbf{V}$$

$$\text{Attention}(\mathbf{Q}, \mathbf{K}, \mathbf{V}) = \text{softmax}\left(\frac{\mathbf{Q}\mathbf{K}^T}{\sqrt{d_k}}\right)\mathbf{V}$$

In the transformer model, this attention computation is repeated across $H$ heads to increase expressivity. Each attention head is calculated by applying the attention function to the transformed input using different sets of weight matrices.

$$\text{head}_i = \text{Attention}(\mathbf{X}\mathbf{W}_{\mathbf{Q_i}}, \mathbf{X}\mathbf{W}_{\mathbf{K_i}}, \mathbf{X}\mathbf{W}_{\mathbf{V_i}});$$

Finally, the outputs from all heads are concatenated and transformed with a linear projection to produce the multi-head attention (MHA) output.

$$\text{MHA}(\mathbf{X}) = \text{Concat}(\text{head}_1, \ldots, \text{head}_h)\mathbf{W}_\mathbf{O}$$

## 3.2 Attention Short-Circuiting

We now describe our approach for isolating the contribution of various attention blocks. The key component of the attention mechanism is the attention weights - computed as an inner product between query and key vectors, *i.e.,*

$$\text{AttentionWeight} = QK^T$$

This enables the model to effectively model both short and long-range dependencies and identify patterns in the input. Pruning methods explore removing the last $K$ layers of the LLM, or sparsifying model weights, while *maintaining* the original output distribution. However, we wish to disable *intermediate* attention modules, and inspect its effect on memorization and general capabilities of the model. While a naive approach would be to simply replace the multi-head self-attention operation in Eq. 1 with **zero**, this has an undesirable effect of changing the distribution of $\mathbf{X}'$ and eventually leading to model collapse, *i.e.,* model generating gibberish text. Instead, we systematically "short-circuit" certain attention layers of the model by replacing this attention weight with the identity matrix *in all heads* of the layer *i.e.,*

$$\text{SHORTCIRCUITATTENTION}(Q, K, V) = \mathbf{I} \cdot \mathbf{V}$$

where $\mathbf{I}$ is the identity matrix, effectively removing the contribution of that attention layer while not destroying the distribution of $\mathbf{X}'$. We perform this 'short-circuit' operation to all attention heads of a layer when applying to any block in the model.

## 3.3 Theoretical Analysis

In this section, we derive the impact of bypassing the attention module of a given block/layer on subsequent layers in the transformer model. We use

the terms block and layer interchangeably. Here, we are specifically interested in the representation of the last token of a given sequence, as the next token is predicted based on its value. First, we derive the effect of short-circuiting attention at a single layer and study the difference in the output representations. We then use the resulting expression to derive the effect of short-circuiting the attention block at a given layer $L$ vs. a later layer $L+1$. Our results indicate that the difference in output representations is amplified as more layers are stacked on top of the short-circuited block, resulting in heavy degradation in generation abilities.

**Notations.** Let $\mathbf{V}^{L-1}$ denote the output representation at layer $L-1$ of the last token in a given sequence $\mathbf{X} = \{x_1, x_2, \ldots, x_n\}$, where $n$ is the length of the input sequence and $\mathbf{V}^{L-1} = \{v_1^{L-1}, v_2^{L-1}, \ldots, v_n^{L-1}\}$ serves as an input to layer $L$. Moving forward, we denote $v^l = v_n^l$ as the representation of the last token for a given layer $l$ for mathematical brevity. The output of layer $L$ with standard attention is represented as $v^L$, while $v_{\text{IA}}^L$ refers to the output of layer $L$ with identity attention. The difference at layer $L$ is denoted by $D^L = v_{\text{IA}}^L - v^L$. Similarly, $v^{L+1}$ represents the output of layer $L+1$ with standard attention. Now, the outputs at layer $L+1$ with attention replaced at layer $L$ and $L+1$ are given by $v_{\text{IA, L}}^{L+1}$ and $v_{\text{IA, L+1}}^{L+1}$, respectively. The output difference at layer $L+1$ when attention is replaced at layer $L$ is denoted by $D_{\text{IA,L}}^{L+1} = v_{\text{IA,L}}^{L+1} - v^{L+1}$, and when attention is replaced at layer $L+1$, it is represented by $D_{\text{IA,L+1}}^{L+1} = v_{\text{IA,L+1}}^{L+1} - v^{L+1}$. The attention weights at layer $L$ are denoted by $\alpha_i^L$, satisfying $\sum_i \alpha_i^L = 1$, while $\mathbf{W}^L$ is the weight matrix of a linear approximation of the feed-forward network (FFN) at layer $L$, and $\epsilon^L$ represents the approximation error at layer $L$. Such linear approximations can be achieved using methods such as Neural Tangent Kernels or piecewise approximations.

**Theorem 1** (Bounding the Difference Between Identity Attention and Standard Attention for a Transformer with a single block). *The normed difference between the output vectors obtained by using standard attention and short-circuited attention for a single transformer block is upper bounded by:*

$$\|v_{IA} - v\| \le (1 + \|\mathbf{W}\|) M (1 - \alpha_l) + \|\epsilon_{IA} - \epsilon\|,$$

*where $M = \max_{i \neq l} \|x_n - x_i\|$, $\mathbf{W}$ is the weight matrix of the FFN layer, and $\epsilon$ is the error in the linear approximation of the activation function.*

*Proof Sketch.* The proof relies on the insight that the value vectors undergo simple MLP-based transformations when passing through the transformer layers, with the attention module replacing each value vector with a convex combination of all input value vectors. Our intervention replaces the convex combination of the last token of the sequence with the identity function. Note that this is a special case of an arbitrary convex combination, where all weight is put on the coefficient of the final token. The proof then follows through when expanding the expressions for the output value vectors with and without short-circuiting the attention, and adding the residual components. The proof has been provided in Appendix A.1. The theorem gives an upper bound of the difference between the outputs of a standard transformer block and a transformer block with short-circuited attention. $\qquad\square$

Next, we use the above bound to derive how this difference propagates if another transformer block is added on top of the short-circuited block and compare it with short-circuiting the attention of the later block alone.

**Theorem 2** (Impact of Replacing Attention in any two consecutive Transformer Layers). *For any two adjacent layers $L$ and $L+1$, the difference in the output representations can be bounded as:*
*When replacing attention at layer $L$:*
$\|D_{IA,L}^{L+1}\| \le (1 + \|W^{L+1}\|)(1 + \alpha_l^{L+1})\|D^L\|$,
*where $\|D^L\| \le (1 + \|W^L\|)M^L(1 - \alpha_l^L) + \|\epsilon_{IA}^L - \epsilon^L\|$.*
*When replacing attention at layer $L+1$:*
$\|D_{IA,L+1}^{L+1}\| \le (1 + \|W^{L+1}\|)M^{L+1}(1 - \alpha_l^{L+1}) + \|\epsilon_{IA}^{L+1} - \epsilon^{L+1}\|$

*Proof Sketch.* This proof uses the result from Theorem 1 to compute the expression of the output of layer $L$ with and without short-circuiting the attention module. Next, we derive the output of layer $L+1$ based on the output of the first and compute the difference in output vectors when replacing the attention in the earlier layer vs. the later layer. The bounds indicate that replacing attention at an earlier layer $L$ introduces differences that propagate and potentially amplify through subsequent layers due to the operations of the FFNs and residual connections. The amplification is influenced by the operator norms of the weight matrices and the attention weights. The theoretical bounds suggest that replacing attention at a lower layer ($L$) can

lead to larger differences at layer $L + 1$, while replacing attention at layer $L + 1$ introduces more localized differences. The complete proof has been provided in Appendix A.1. □

## 4 Experiments

Using our model attribution method described in Sec. 3, we now study the effect of 'short-circuiting' the attention mechanism in various blocks of an LLM on its memorization of the training dataset.

### 4.1 Experimental Setup

**Models.** We consider six LLMs of varying scales across two architectures – GPTNeo{1.3B, 2.7B} (Black et al., 2021) and Pythia{1.4B, 2.8B, 6.9B, 12B} (Biderman et al., 2023), trained on the publicly available Pile dataset (Gao et al., 2020), allowing us to find samples from the training corpus that show extractable memorization.

**Memorized Dataset.** For each model family, we collect 15k samples from the training dataset which are highly memorized (more than 90% samples show *extractable memorization*) by all model scales. These samples were made available by Carlini et al. (2020); Prashanth et al. (2024) in their works on investigating memorization in LLMs. The memorized samples collected for GPTNeo models have a prefix length $l_p$ of 150 tokens and a suffix length $l_s$ of 50 tokens. The memorized samples collected for Pythia models have a prefix length $l_p$ of 32 tokens and a suffix length of $l_s$ of 32 tokens.

**Evaluation Benchmarks.** Across our model attribution experiments, we also aim to quantify the general capabilities of the model alongside the memorization characteristics. To this end, we employ five different benchmarks – ARC (Clark et al., 2018), HellaSwag (Zellers et al., 2019), LAMBADA (Paperno et al., 2016), PIQA (Bisk et al., 2020), and Wikitext (Merity et al., 2022) – which test the model's performance on a variety of capabilities. In particular, we include benchmarks that test the reasoning and language understanding of the LLM. Refer to Appendix A.2 for a detailed description of each benchmark.

**Memorization Metrics.** The focus of this work is to study the effect of various model components on the memorization characteristics of the LLM. While exact memorization, as defined in the previous section, is the strongest indicator of memorization, we also employ two additional metrics of memorization, for a total of three:

*1) Exact Match:* This is simply the definition of extractable memorization, realized as a metric, *i.e.,*

$$\text{EM}(f_\theta, p, s) = \mathbf{1}(\text{GS}(f_\theta, p) == s)) \quad (3)$$

*2) Token Accuracy:* Samples may not be verbatim memorized, but still show a large overlap with the exact training example. To account for this, we calculate the number of matching tokens between the generated sequence, and the suffix, *i.e.,*

$$\text{TA}(f_\theta, p, s) = \frac{1}{l_s} \sum_{i=1}^{l_s} (\text{GS}(f_\theta, p)[i] == s[i]))$$
$$(4)$$

We cut off the generation process after $l_s$ tokens and ignore any further tokens.

*3) Completion Entropy:* is the total entropy of the logit distribution for each token of the suffix conditioned on the prefix for memorized and non-memorized samples. Let the concatenated sequence $(p||s)$ be represented by a sequence of tokens $(x_1, \ldots, x_{l_p+l_s})$. The completion entropy (CE) is defined as follows

$$\text{CE}(f_\theta, p, s) = -\sum_{i=l_p}^{l_p+l_s-1} \sum_{j=1}^{|V|} p_\theta^j(x_{i+1}|x_{1:i}) \log p_\theta^j(x_{i+1}|x_{1:i}),$$
$$(5)$$

where $p_\theta^j$ is the prediction probability for the $j^{\text{th}}$ token in the vocabulary from the LLM $f_\theta$ and $|V|$ is the vocabulary size.

### 4.2 Results

Next, we present the insights from our experiments.

#### 4.2.1 Memorization vs. Performance

We aim to understand the trade-off between the memorization and downstream performance of LLMs. In particular, we strive to answer: "*Can we isolate model components that correspond to general performance and memorization, or are the two deeply intertwined?*" To answer this question, we produce $L$ edited variants of the given LLM: $\{f_\theta^1, f_\theta^2, \ldots, f_\theta^L\}$ by short-circuiting the attention mechanism in each of the $L$ blocks of the model. Next, we quantify the memorization performance of each model variant using the memorization metrics defined in Sec. 3 and their performance on standard benchmark. Following Prashanth et al. (2024), we evaluate all memorization metrics using greedy sampling and leave the exploration of other targeted extraction techniques as future work. We present our findings in Fig. 1 for the GPTNeo-1.3B model across all benchmarks. Interestingly, we observe that short-circuiting attention in many

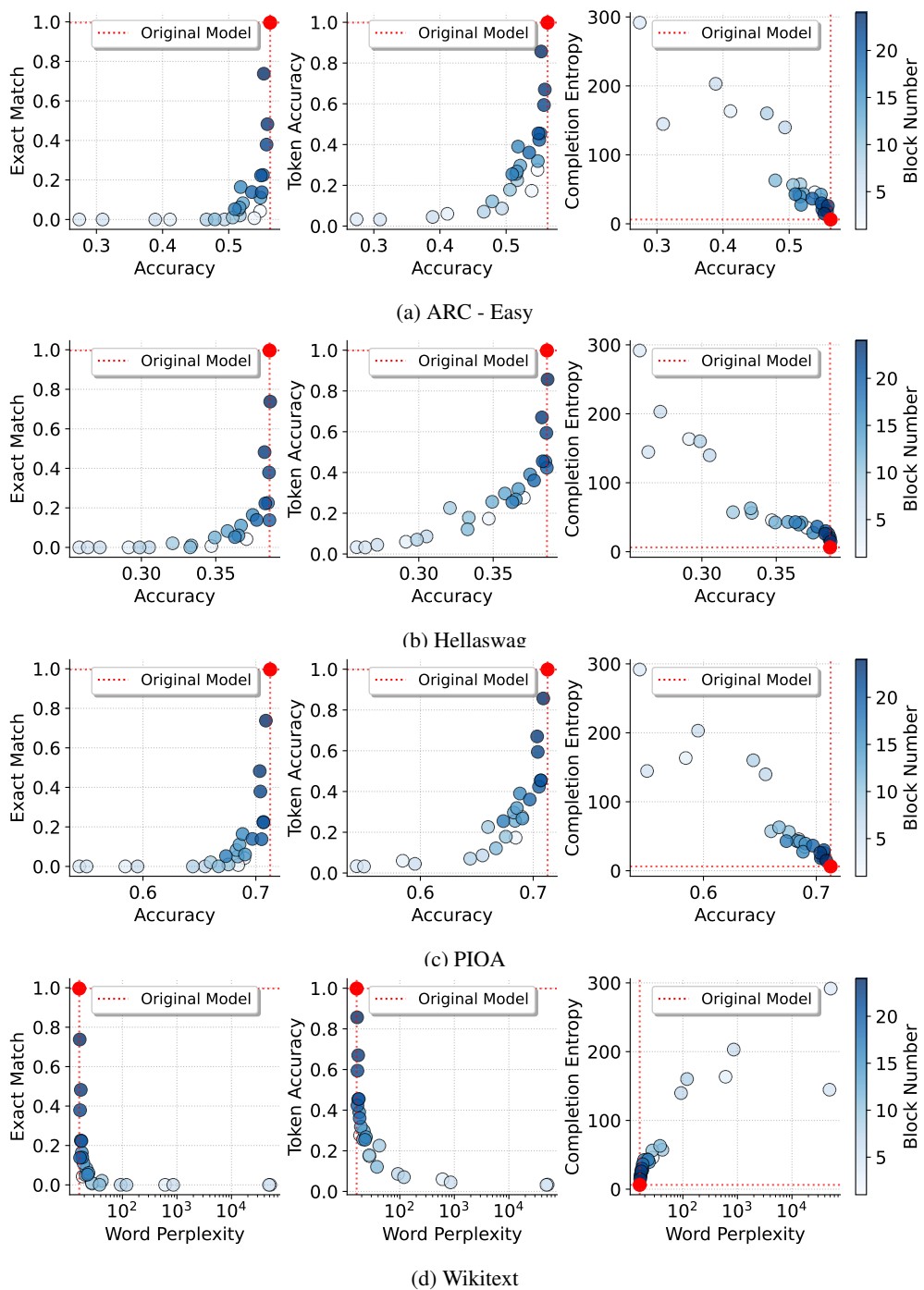

Figure 1: Memorization vs. Downstream Performance for GPTNeo-1.3B with short-circuiting applied to the attention mechanism of each block independently. When applied to later blocks of the model, attention short-circuiting consistently yields lower memorization scores, while maintaining downstream benchmark scores. Results on all models and datasets in Appendix A.3.

blocks (particularly, the last quartile blocks, *i.e.,* layers 18-24 for GPTNeo-1.3B) leads to a significant drop in memorization, with negligible drop in model performance. In some cases, the short-circuited variants even outperform the benchmark performance of the original model (see LAMBADA results in Fig. A8c). Further, short-circuiting the attention mechanism in the earlier layers of the LLMs leads to almost no memorization and leads to a significant drop of benchmark performance, with accuracy no better than random chance, *i.e.,* the LLM collapses. We observe that applying attention short-circuiting in the first block (pure white circle in Fig. 1) sometimes spuriously leads to high performance, however the model is unable to generate any coherent sequences in this

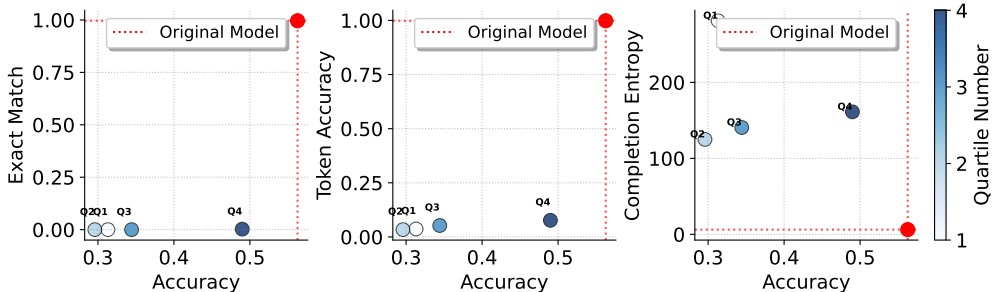

Figure 2: Memorization vs. Downstream Performance on the PIQA benchmark for GTPNeo-1.3B with short-circuiting applied to all attention blocks in each quartile of the model layers. Short-circuiting multiple layers shows a significant drop in memorization, and short-circuiting the last quartile still maintains a large portion of performance on downstream tasks. Additional results on other models and benchmarks can be found in Appendix A.3.

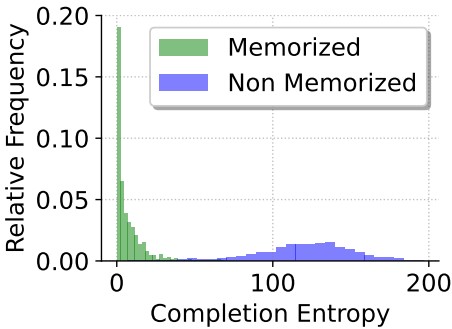

Figure 3: Completion Entropy Scores for Memorized and Non Memorized Samples. Model used is GPTNeo-1.3B. Memorized samples consistently show significantly lower completion entropy scores.

state, as we discuss in Sec. 4.2.4. Please refer to Appendix A.3 for additional results on other LLMs and benchmarks.

### 4.2.2 Short-Circuiting Multiple Layers

Our experiments in Sec. 4.2.1 mainly focused on the situation wherein short-circuiting is applied to a single attention block of an LLM. As discussed previously, short-circuiting applied to the attention mechanism in the last quartile yields a significant drop in memorization without significantly affecting downstream performance, motivating further investigation by short-circuiting **groups** of attention blocks, which we demonstrate in Fig. 2. Please refer to Appendix A.3 for additional results on other LLMs and benchmarks.

Across all models, applying short-circuiting to all layers in a quartile results in *near-zero* exact match memorization. Additionally, we notice a similar trend as the single-layer experiments, with short-circuiting applied to the layers in the final quartile of the model resulting in the least loss in downstream performance. This is an encouraging finding, demonstrating the absence of verbatim

memorization while retaining most of the original performance on downstream tasks.

### 4.2.3 Short-Circuiting across Model Scale

Our findings in Sec. 4.2.1 show that the attention in later blocks of LLMs contributes strongly to memorization and has an insignificant effect on the downstream performance. This insight is crucial within the context of memorization in LLMs as it **implies that the attention mechanism in later layers of a transformer can be skipped during inference**, with the desirable result of lower memorization and retaining downstream performance. Next, we investigate whether this insight generalizes to LLMs of different scale (increasing parameter size), and our findings in Fig. 4 show a consistent trend across four different scales of the Pythia models, wherein short-circuiting attention in later blocks of the LLM results in reduced memorization. We note that larger models are more resistant to this phenomenon, with the gap in memorization between edited and original models reducing as model scale increases. We hypothesize that applying attention short-circuiting to groups of blocks in parallel may result in a larger drop in memorization scores, though we leave this for future exploration.

### 4.2.4 Qualitative Analysis

Here, we analyze the qualitative effect of short-circuiting attention on its generation performance. In Fig. 5, we show the result of short circuiting the attention mechanism in four different blocks of GPTNeo-1.3B, finding that short-circuiting attention in the earlier blocks leads to a breakdown of coherent generations. We hypothesize that the earlier layers of the LLM are responsible for general semantics and grammar and any change in them results in model collapse. The same effect

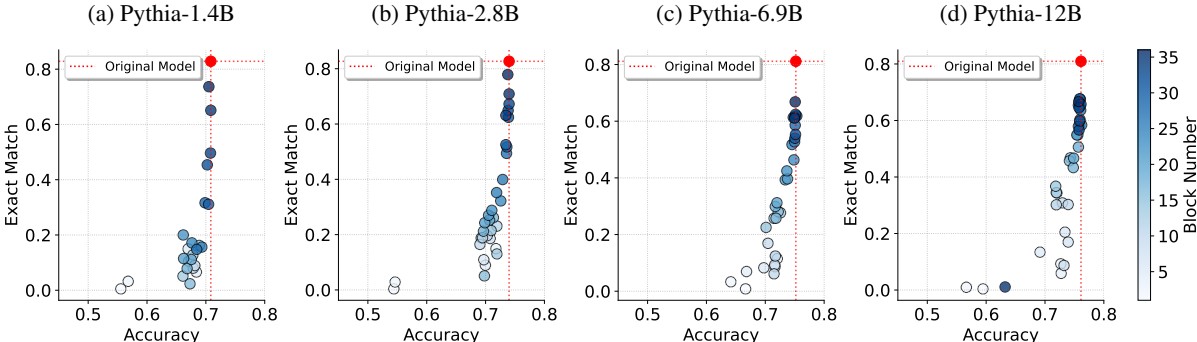

Figure 4: Results for short-circuiting the attention mechanism in each block of Pythia models across four scales - 1.4B, 2.8B, 6.9B, 12B. Short-circuiting in later blocks of the LLM consistently shows lower memorization with a negligible drop in downstream benchmark performance on PIQA across all model scales. Additional results on other benchmarks can be found in the Appendix A.3.

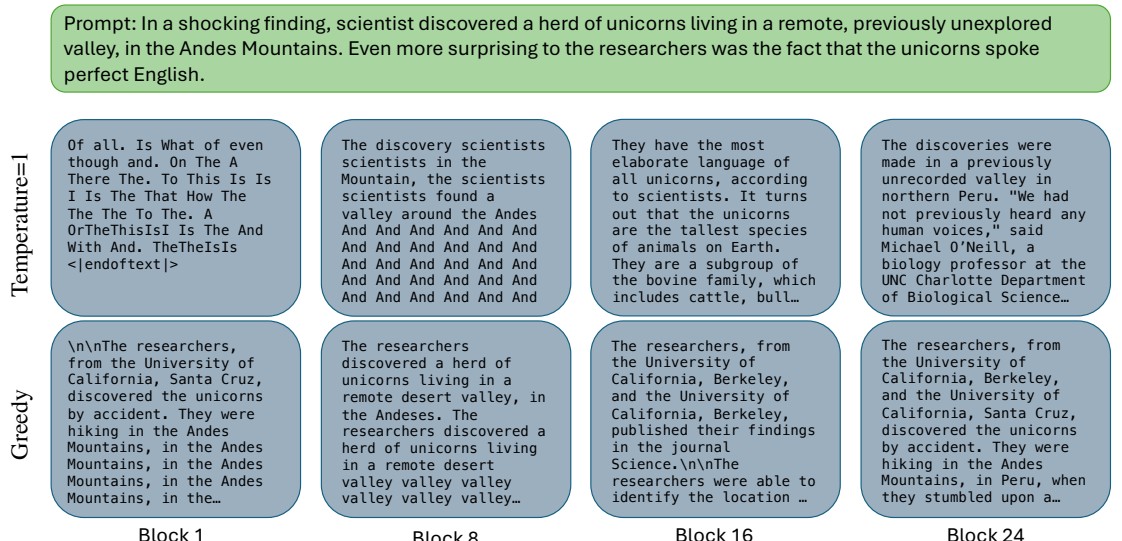

Figure 5: Samples generated using greedy and temperature sampling by GPTNeo-1.3B when short-circuiting attention at different blocks. Attention short-circuiting in earlier blocks leads to model collapse, while applying it to later blocks still results in coherent generations. Query prompt is taken from the GPT-2 paper (Radford et al., 2019).

is not observed when short-circuiting later blocks of the LLM, wherein the model still generated coherent and plausible completions to the given prompt. This behavior is also reflected in the Wikitext perplexity scores obtained by short-circuiting each of the different attention blocks in the model, shown in Fig. 6, where the short-circuit operation when applied to later attention blocks, does not degrade the perplexity scores significantly.

### 4.2.5 Reasoning vs. Non-reasoning Tasks

In the above sections, we evaluate short-circuited models across different scales on a comprehensive set of benchmarks, where results clearly show that short-circuiting applied to the later blocks of an LLM yields a huge decrease in all memorization metrics, without sacrificing model performance significantly. Here, we further investigate this phe-

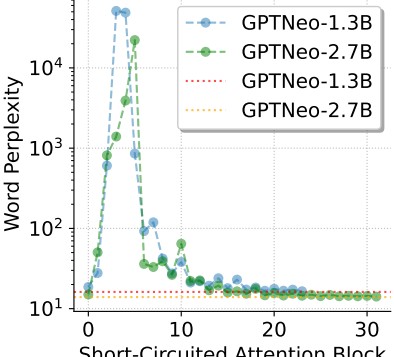

Figure 6: Wikitext word perplexity scores for GPTNeo models with attention short-circuiting applied to each block. Short-circuiting attention in later blocks results in minimal degradation in perplexity scores. Dashed lines indicate vanilla model performance.

nomenon by dividing the benchmark tasks into reasoning and language understanding tasks. We measure the average drop in performance across bench-

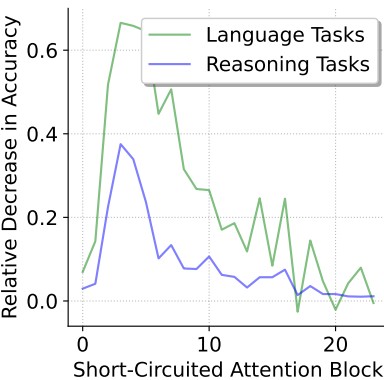

Figure 7: Relative decrease in benchmark performance for reasoning and language understanding tasks after short-circuiting attention mechanism in each block of GPTNeo-1.3B. Reasoning tasks are more resistant to attention short-circuiting, especially in later blocks. Results for additional models in Appendix A.3.

marks in each type of task after short-circuiting the attention mechanism in each layer.

Results in Fig. 7 show a clear pattern, where the average relative decrease in score on reasoning tasks (PIQA, ARC-Easy) is much lesser than in the case of language understanding tasks (Hellaswag, LAMBADA). Additionally, a large set of transformer blocks show almost **zero** drop in relative performance on reasoning tasks after attention short-circuiting, suggesting that these blocks contribute almost exclusively to memorization, while not improving the reasoning capabilities of the model. We present consistent results across all six models in Appendix A.3.

## 5    Conclusion

Our study explores how attention modules at different depths contribute to memorization in LLMs. By systematically bypassing attention modules at targeted blocks while preserving other components, we isolate their impact on memorization and generalization. Our theoretical analysis showed that bypassing attention in earlier layers disrupts output representations, harming performance and generalization, while interventions in deeper layers prevent memorized content generation without impairing generalization abilities. Empirical experiments with Pythia and GPT-Neo models aligned with these predictions, indicating that deeper attention blocks play a critical role in memorization phenomena. This work enhances understanding of how transformer components influence memorization and offers a practical approach to address ethical concerns like privacy violations. By reducing memorization without degrading model performance, we contribute to developing more ethical and reliable LLMs suitable for real-world applications. Our targeted intervention strategy opens avenues for future research into architectural modifications that balance memorization and generalization, advancing responsible artificial intelligence technologies.

## 6    Limitations and Future Directions

While our approach shows promising results, several limitations warrant further investigation:

1. Effects of attribution on multiple attention blocks: Our work provides the first step towards analyzing memorization from an architectural perspective. We study memorization by isolating and short-circuiting individual attention modules. The current work does not include studying the effects of short-circuiting multiple attention modules simultaneously, which is an interesting direction for future work.

2. Effects of non-attention modules: Additionally, the work currently studies the effects of only attention modules, not the other transformer components such as layer normalizations or MLP layers. The study of these components is also another interesting direction for future work.

3. Study on targeted extraction: While our work shows that attention short-circuiting is successful in reducing verbatim memorization, we only study this under the conditions of greedy sampling. An extension of this work to include more targeted extraction attacks is an important future exploration.

4. Study of models trained on closed datasets: Finally, the models and datasets studied in this work are on the smaller side compared to commercial models like GPT-4. However, such large-scale models cannot be directly evaluated for memorization as the data used to train such models is not publicly available. However, our results indicate that applying the proposed interventions can nevertheless prevent them from generating memorized content, without compromising on their generation capabilities.

## 7 Ethical Considerations

Our method studies the impact of attention modules on memorization and provides a mechanism to overcome generation of memorized sequences. All data used in our studies come from the public domain and do not contain private information. All models are also publicly available. Hence, our work does not have any ethical concerns that need to be explicitly addressed.

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

# A  Appendix

## Contents

## A.1   Proofs

**Theorem 1** (Bounding the Difference Between Identity Attention and Standard Attention for a Transformer with a single block). *The normed difference between the output vectors obtained by using standard attention and short-circuited attention for a single transformer block is upper bounded by:*

$$\|v_{IA} - v\| \le (1 + \|\mathbf{W}\|)M(1 - \alpha_l) + \|\epsilon_{IA} - \epsilon\|,$$

*where $M = \max_{i \ne l} \|x_n - x_i\|$, $\mathbf{W}$ is the weight matrix of the FFN layer, and $\epsilon$ is the error in the linear approximation of the activation function.*

*Proof.* We begin by expressing the difference $D$ between the output value vectors after the application of identity attention and standard attention:

$$D = v^{IA} - v$$

Substituting the expressions for $v^{IA}$ and $v$:

$$D = [2x_n + 2\,\text{FFN}(x_n) + \epsilon^{IA}] - [z_n + \text{FFN}(z_n) + \epsilon]$$

where $z_n$ is the output of the standard attention module corresponding to the input $x_n$, after the residual addition.

Simplifying, we get:

$$D = [2x_n - z_n] + [2\,\text{FFN}(x_n) - \text{FFN}(z_n)] + (\epsilon^{IA} - \epsilon)$$

The first term is:

$$2x_n - z_n = 2x_n - \left(\sum_i \alpha_i x_i + x_n\right) = x_n - \sum_i \alpha_i x_i$$

This can be rewritten as:

$$2x_n - z_n = (1 - \alpha_n)x_n - \sum_{i \ne n} \alpha_i x_i$$

Since $\alpha_n + \sum_{i \ne n} \alpha_i = 1$, we have:

$$2x_n - z_n = \sum_{i \ne n} \alpha_i (x_n - x_i)$$

Now consider the second term:

$$2\,\text{FFN}(x_n) - \text{FFN}(z_n) = 2\,\text{FFN}(x_n) - \text{FFN}\left(\sum_i \alpha_i x_i + x_n\right)$$

Using the linearity of the FFN:

$$2\,\text{FFN}(x_n) - \text{FFN}(z_n) = \text{FFN}(x_n) - \sum_i \alpha_i \text{FFN}(x_i)$$

Thus, the total difference $D$ becomes:

$$D = \sum_{i \neq n} \alpha_i \left( x_n - x_i + \text{FFN}(x_n) - \text{FFN}(x_i) \right) + (\epsilon^{IA} - \epsilon)$$

Since $\text{FFN}(x_n) - \text{FFN}(x_i) = \text{FFN}(x_n - x_i)$ due to the linearity of FFN, we can rewrite $D$ as:

$$D = \sum_{i \neq n} \alpha_i \left( \delta x_i + \text{FFN}(\delta x_i) \right) + (\epsilon^{IA} - \epsilon)$$

where $\delta x_i = x_n - x_i$.
Now, using the triangle inequality:

$$\|D\| \leq \sum_{i \neq n} \alpha_i \left( \|\delta x_i\| + \|\text{FFN}(\delta x_i)\| \right) + \|\epsilon^{IA} - \epsilon\|$$

Since $\|\text{FFN}(\delta x_i)\| = \|W \delta x_i\| \leq \|W\| \|\delta x_i\|$, we have:

$$\|D\| \leq \sum_{i \neq n} \alpha_i \left( \|\delta x_i\| + \|W\| \|\delta x_i\| \right) + \|\epsilon^{IA} - \epsilon\|$$

Factoring $\|\delta x_i\|$:

$$\|D\| \leq (1 + \|W\|) \sum_{i \neq n} \alpha_i \|\delta x_i\| + \|\epsilon^{IA} - \epsilon\|$$

Let $M = \max_{i \neq n} \|x_n - x_i\|$. Then:

$$\|\delta x_i\| \leq M$$

Thus, we can bound $D$ as:

$$\|D\| \leq (1 + \|W\|) M \sum_{i \neq n} \alpha_i + \|\epsilon^{IA} - \epsilon\|$$

Since $\sum_{i \neq n} \alpha_i = 1 - \alpha_n$, we obtain the final bound:

$$\|D\| \leq (1 + \|W\|) M (1 - \alpha_n) + \|\epsilon^{IA} - \epsilon\|$$

$\square$

**Theorem 2** (Impact of Replacing Attention in any two consecutive Transformer Layers). *For any two adjacent layers $L$ and $L + 1$, the difference in the output representations can be bounded as:*
*When replacing attention at layer $L$:*
$\|D_{IA,L}^{L+1}\| \leq (1 + \|W^{L+1}\|)(1 + \alpha_n^{L+1})\|D^L\|$, *where* $\|D^L\| \leq (1 + \|W^L\|)M^L(1 - \alpha_n^L) + \|\epsilon_{IA}^L - \epsilon^L\|$.
*When replacing attention at layer $L + 1$:*
$\|D_{IA,L+1}^{L+1}\| \leq (1 + \|W^{L+1}\|)M^{L+1}(1 - \alpha_l^{L+1}) + \|\epsilon_{IA}^{L+1} - \epsilon^{L+1}\|$

*Proof.* **Case 1:** Replacing Attention at Layer $L$
  Standard Attention Output at Layer $L$:

$$v^L = z^L + \text{FFN}^L(z^L) + \epsilon^L,$$

where

$$z^L = \sum_i \alpha_i^L v_i^{L-1} + v_n^{L-1}.$$

Identity Attention Output at Layer $L$:

$$v_{\text{IA}}^L = z_{\text{IA}}^L + \text{FFN}^L(z_{\text{IA}}^L) + \epsilon_{\text{IA}}^L,$$

where

$$z_{\text{IA}}^L = 2v_n^{L-1}.$$

Difference at Layer $L$:

$$D^L = v_{\text{IA}}^L - v^L = \delta z^L + \text{FFN}^L(\delta z^L) + (\epsilon_{\text{IA}}^L - \epsilon^L),$$

where

$$\delta z^L = z_{\text{IA}}^L - z^L = \sum_{i \neq n} \alpha_i^L(v_n^{L-1} - v_i^{L-1}).$$

**Bounding $\|D^L\|$:**
Using the triangle inequality and linearity of the FFN approximation:

$$\|D^L\| \leq (1 + \|\mathbf{W}^L\|)\|\delta z^L\| + \|\epsilon_{\text{IA}}^L - \epsilon^L\|.$$

Since

$$\|\delta z^L\| \leq M^L(1 - \alpha_n^L),$$

where $M^L = \max_{i \neq n} \|v_n^{L-1} - v_i^{L-1}\|$, we have:

$$\|D^L\| \leq (1 + \|\mathbf{W}^L\|)M^L(1 - \alpha_n^L) + \|\epsilon_{\text{IA}}^L - \epsilon^L\|.$$

**Propagating the Difference to Layer $L + 1$:**
The modified input to layer $L + 1$ is:

$$v_{\text{IA}}^L = v^L + D^L.$$

Standard Attention Output at Layer $L + 1$:

$$v^{L+1} = z^{L+1} + \text{FFN}^{L+1}(z^{L+1}) + \epsilon^{L+1},$$

where

$$z^{L+1} = \sum_i \alpha_i^{L+1} v_i^L + v_n^L.$$

Modified Attention Output at Layer $L + 1$:
Since the inputs are modified only through $v_{\text{IA}}^L$, the attention output becomes:

$$z_{\text{IA,L}}^{L+1} = z^{L+1} + \alpha_n^{L+1} D^L + D^L.$$

Difference at Layer $L + 1$:

$$D_{\text{IA,L}}^{L+1} = v_{\text{IA,L}}^{L+1} - v^{L+1} = \delta z^{L+1} + \text{FFN}^{L+1}(\delta z^{L+1}) + (\epsilon_{\text{IA,L}}^{L+1} - \epsilon^{L+1}),$$

where

$$\delta z^{L+1} = z_{\text{IA,L}}^{L+1} - z^{L+1} = (1 + \alpha_n^{L+1})D^L.$$

Bounding $\|D_{\text{IA,L}}^{L+1}\|$:
Using the triangle inequality and linearity:

$$\|D_{\text{IA,L}}^{L+1}\| \leq (1 + \|\mathbf{W}^{L+1}\|)\|\delta z^{L+1}\| + \|\epsilon_{\text{IA,L}}^{L+1} - \epsilon^{L+1}\|.$$

Substituting $\delta z^{L+1} = (1 + \alpha_n^{L+1})D^L$:

$$\|D_{\text{IA,L}}^{L+1}\| \leq (1 + \|\mathbf{W}^{L+1}\|)(1 + \alpha_n^{L+1})\|D^L\| + \|\epsilon_{\text{IA,L}}^{L+1} - \epsilon^{L+1}\|.$$

Final Bound for $D_{\text{IA,L}}^{L+1}$:
Combining with the bound for $\|D^L\|$:

$$\|D_{\text{IA,L}}^{L+1}\| \leq (1 + \|\mathbf{W}^{L+1}\|)(1 + \alpha_n^{L+1})\left[(1 + \|\mathbf{W}^L\|)M^L(1 - \alpha_n^L) + \|\epsilon_{\text{IA}}^L - \epsilon^L\|\right] + \|\epsilon_{\text{IA,L}}^{L+1} - \epsilon^{L+1}\|.$$

**Case 2:** Replacing Attention at Layer $L + 1$
Difference at Layer $L + 1$:
Standard Attention Output at Layer $L + 1$:

$$v^{L+1} = z^{L+1} + \text{FFN}^{L+1}(z^{L+1}) + \epsilon^{L+1},$$

where

$$z^{L+1} = \sum_i \alpha_i^{L+1} v_i^L + v_n^L.$$

Identity Attention Output at Layer $L + 1$:

$$v_{\text{IA,L+1}}^{L+1} = z_{\text{IA,L+1}}^{L+1} + \text{FFN}^{L+1}(z_{\text{IA,L+1}}^{L+1}) + \epsilon_{\text{IA}}^{L+1},$$

where

$$z_{\text{IA,L+1}}^{L+1} = 2v_n^L.$$

Difference at Layer $L + 1$:

$$D_{\text{IA,L+1}}^{L+1} = v_{\text{IA,L+1}}^{L+1} - v^{L+1} = \delta z_{\text{IA}}^{L+1} + \text{FFN}^{L+1}(\delta z_{\text{IA}}^{L+1}) + (\epsilon_{\text{IA}}^{L+1} - \epsilon^{L+1}),$$

where

$$\delta z_{\text{IA}}^{L+1} = z_{\text{IA,L+1}}^{L+1} - z^{L+1} = \sum_{i \neq n} \alpha_i^{L+1}(v_n^L - v_i^L) + (v_n^L - v_n^L) = \sum_{i \neq n} \alpha_i^{L+1}(v_n^L - v_i^L).$$

Bounding $\|D_{\text{IA,L+1}}^{L+1}\|$:
Using the triangle inequality and linearity:

$$\|D_{\text{IA,L+1}}^{L+1}\| \leq (1 + \|\mathbf{W}^{L+1}\|)\|\delta z_{\text{IA}}^{L+1}\| + \|\epsilon_{\text{IA}}^{L+1} - \epsilon^{L+1}\|.$$

Since

$$\|\delta z_{\text{IA}}^{L+1}\| \leq M^{L+1}(1 - \alpha_n^{L+1}),$$

where $M^{L+1} = \max_{i \neq n} \|v_n^L - v_i^L\|$, we have:

$$\|D_{\text{IA,L+1}}^{L+1}\| \leq (1 + \|\mathbf{W}^{L+1}\|)M^{L+1}(1 - \alpha_n^{L+1}) + \|\epsilon_{\text{IA}}^{L+1} - \epsilon^{L+1}\|.$$

Replacing attention at layer $L$ introduces differences that propagate and potentially amplify through subsequent layers due to the operations of the FFNs and residual connections. The amplification is influenced by the operator norms of the weight matrices and the attention weights. The theoretical bounds suggest that replacing attention at a lower layer ($L$) can lead to larger differences at layer $L + 1$, while replacing attention at layer $L + 1$ introduces more localized differences.

$\square$

## A.2 Description of Benchmarks

In this section, we provide detailed descriptions of the various benchmarks that we use in our experiments. We use a diverse set of reasoning and language understanding tasks to comprehensively understand the effect of attention short-circuiting on downstream benchmarks.

**Lambada (Paperno et al., 2016).** Evaluates contextual understanding with word prediction tasks, where the model is given a long text passage and asked to predict the last word, requiring deep comprehension.

**AI2 Reasoning Challenge (ARC) (Clark et al., 2018) - Easy.** Measures reasoning with elementary-level science and general knowledge multiple-choice questions from standardized tests that require straightforward answers.

**Hellaswag (Zellers et al., 2019):.** Assesses commonsense reasoning with multiple-choice narrative completion, where a model must select the most logical ending from several options for scenarios like activities or descriptions.

**Wikitext (Merity et al., 2022):.** It contains well-structured and formal writing, focusing on the model's ability to generate coherent text by testing LLMs in predicting text sequences from Wikipedia articles.

**Physical Interaction: Question Answering (PIQA) (Bisk et al., 2020):.** Tests physical commonsense reasoning using multiple-choice questions about everyday tasks, where the LLM must select the more reasonable option between two possible solutions.

## A.3 Additional Results

In this section, we present results for all our experiments in Sec. 4 for additional benchmarks and models, showing consistent trends across all six models (GPTNeo-{1.3B, 2.7B}, Pythia-{1.4B, 2.8B, 6.9B, 12B}) and five benchmarks (ARC-Easy, HellaSwag, LAMBADA, PIQA, and Wikitext).

### A.3.1 Memorization vs Downstream Performance on Additional Models

In figs. A8 to A12, we present additional results comparing the memorization scores and downstream benchmark performance for all remaining models (GPTNeo-2.7B, Pythia-1.4B, 2.8B, 6.9B, 12B). We consistently observe lower memorization scores without a drop in benchmark performance when applying short circuiting to the attention mechanism of later blocks of the LLM, across all models and benchmarks. As mentioned in Sec. 4, we observe that the drop in memorization is less pronounced in larger models, suggesting a deeper investigation by short-circuiting attention mechanisms in **groups** of blocks.

### A.3.2 Applying Short-Circuiting to Multiple Layers

In figs. A13 to A18, we show results of applying attention short-circuiting to **groups** of attention blocks on additional LLMs and benchmarks. We consistently observe that across all quartiles of the model, short-circuiting the attention mechanism in all blocks of the quartile results in *near-zero* verbatim memorization, while short-circuiting applied to the *last quartile* of attention blocks still retains a large portion of capabilities on benchmarks.

### A.3.3 Short-Circuiting across Model Scale on Additional Benchmarks

In figs. 4 and A19 to A21 we show results of attention short-circuiting across model scales on additional benchmarks (HellaSwag, ARC-Easy, LAMBADA, and Wikitext). Our findings remain consistent across benchmarks, with later blocks of all model scales showing a drop in memorization without sacrificing downstream performance when attention short-circuiting is applied.

### A.3.4 Short-Circuiting on Reasoning vs. Non-Reasoning tasks on Additional Models

In figs. A23a to A23e we show results for additional models on the relative drop in performance on benchmarks when short-circuiting the attention mechanism. Our findings remain consistent across all model families and scales, with reasoning tasks displaying a lower relative drop in performance when short-circuiting attention in all blocks.

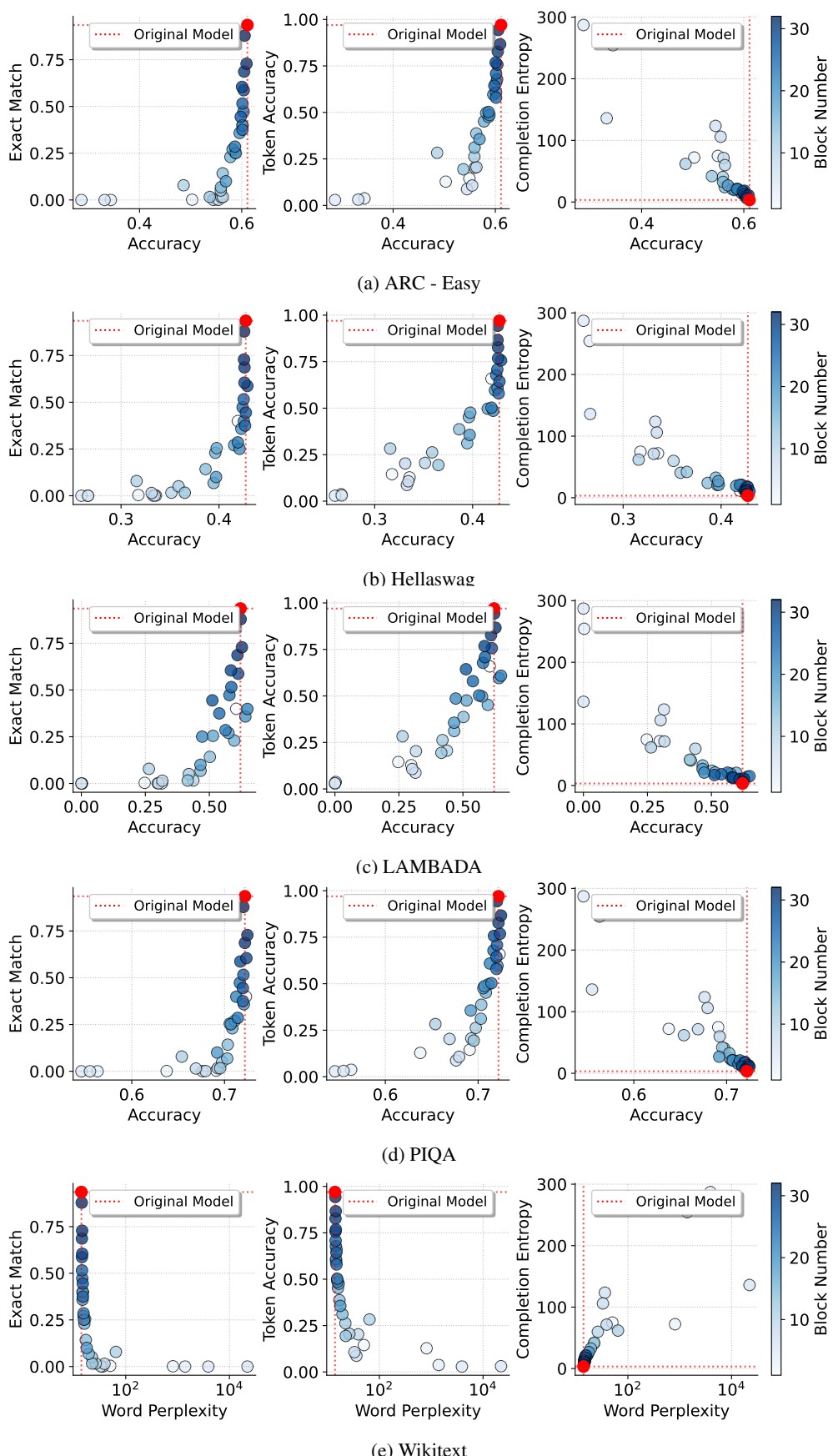

Figure A8: Memorization vs. Downstream Performance for GPTNeo-2.7B with short-circuiting applied to the attention mechanism of each block independently.

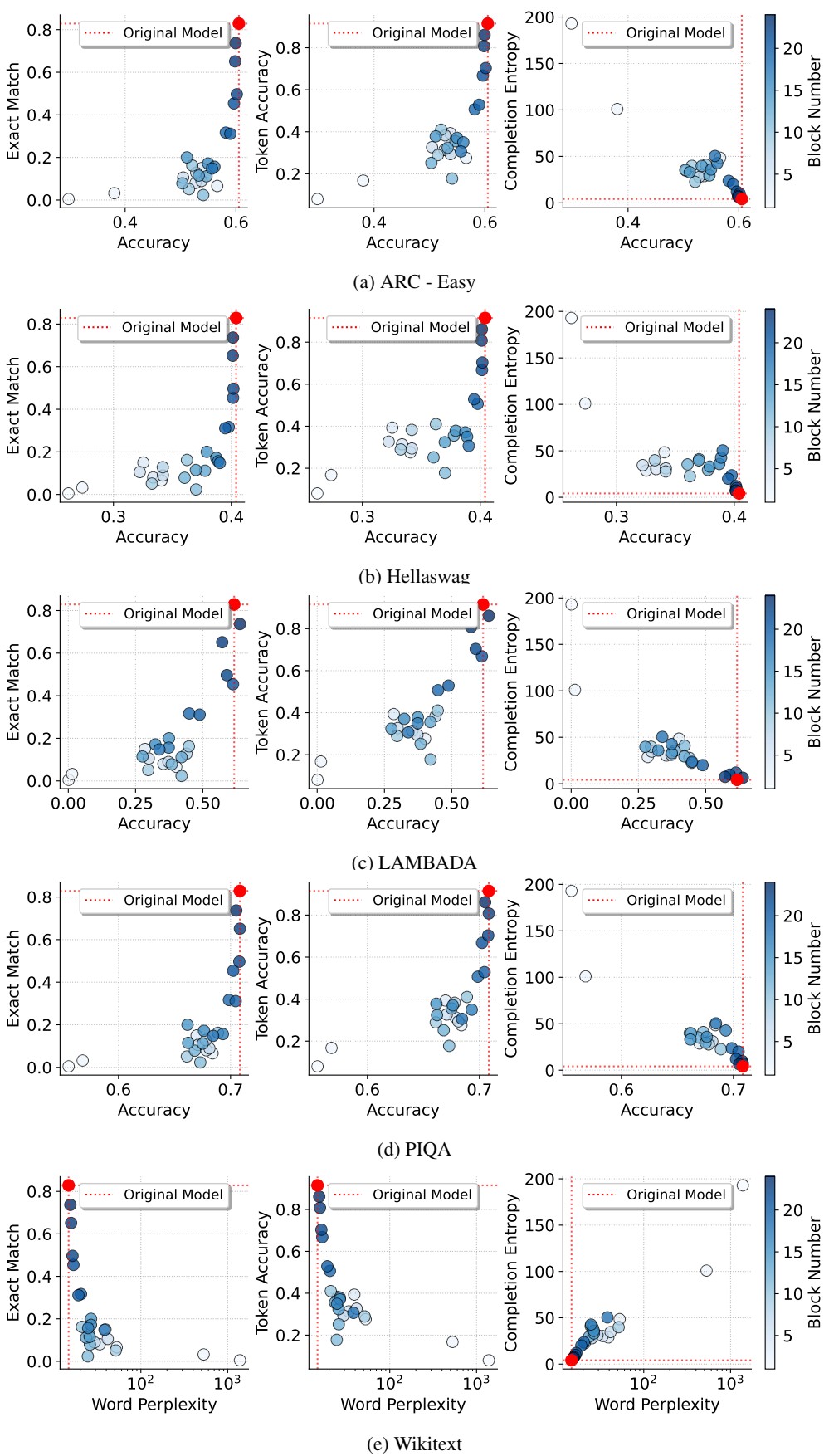

Figure A9: Memorization vs. Downstream Performance for Pythia-1.4B with short-circuiting applied to the attention mechanism of each block independently.

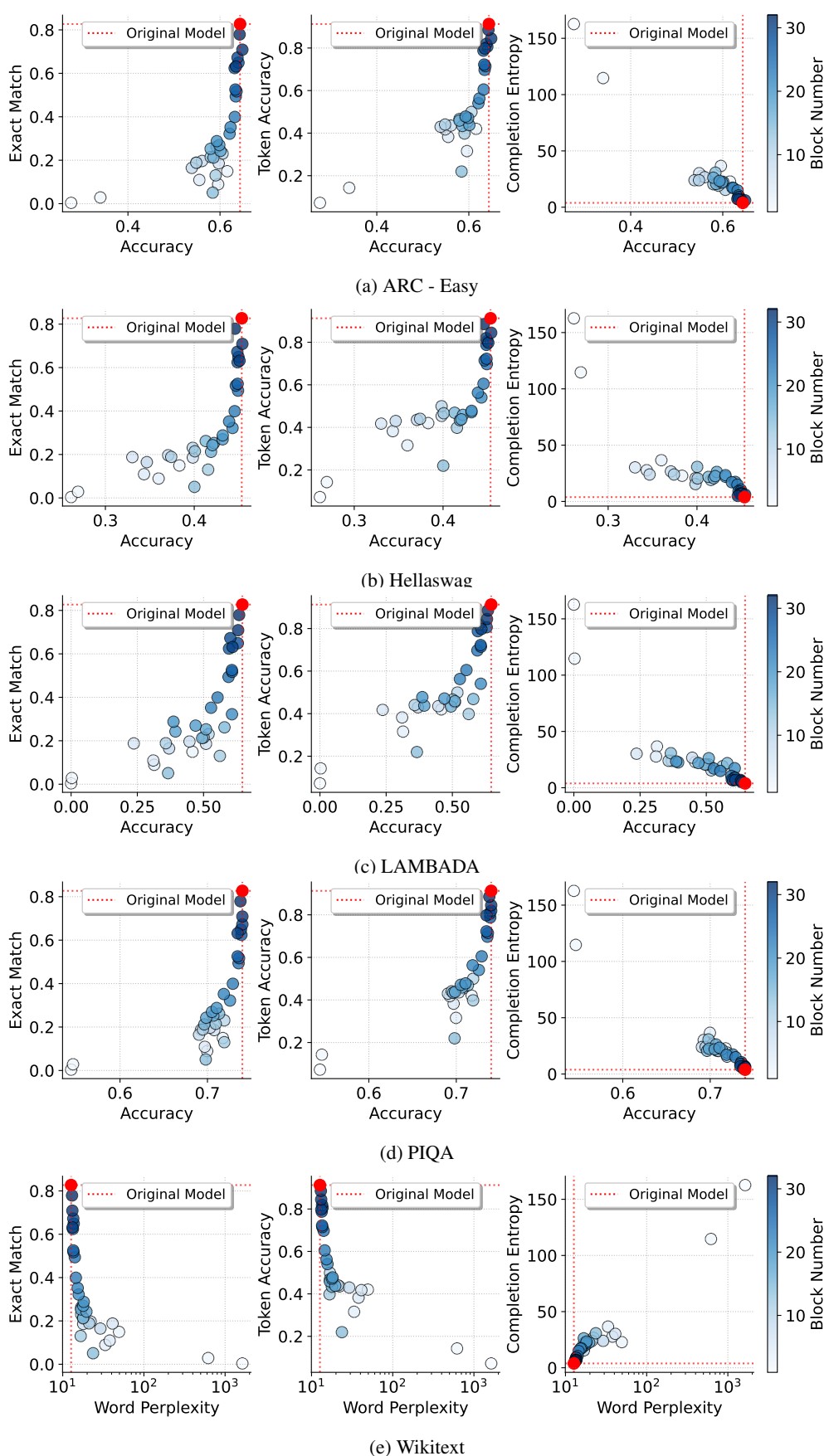

Figure A10: Memorization vs. Downstream Performance for Pythia-2.8B with short-circuiting applied to the attention mechanism of each block independently.

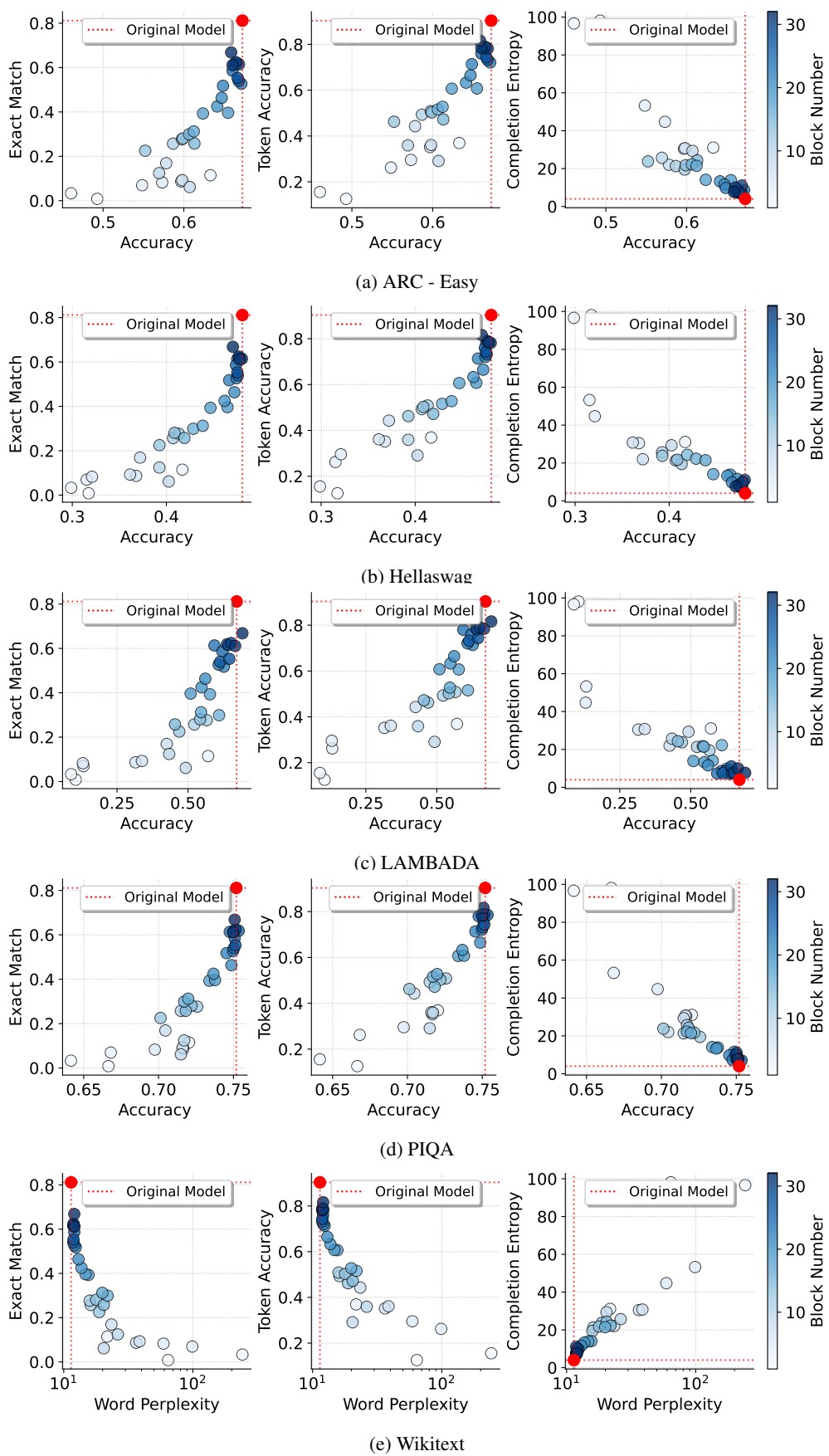

Figure A11: Memorization vs. Downstream Performance for Pythia-6.9B with short-circuiting applied to the attention mechanism of each block independently.

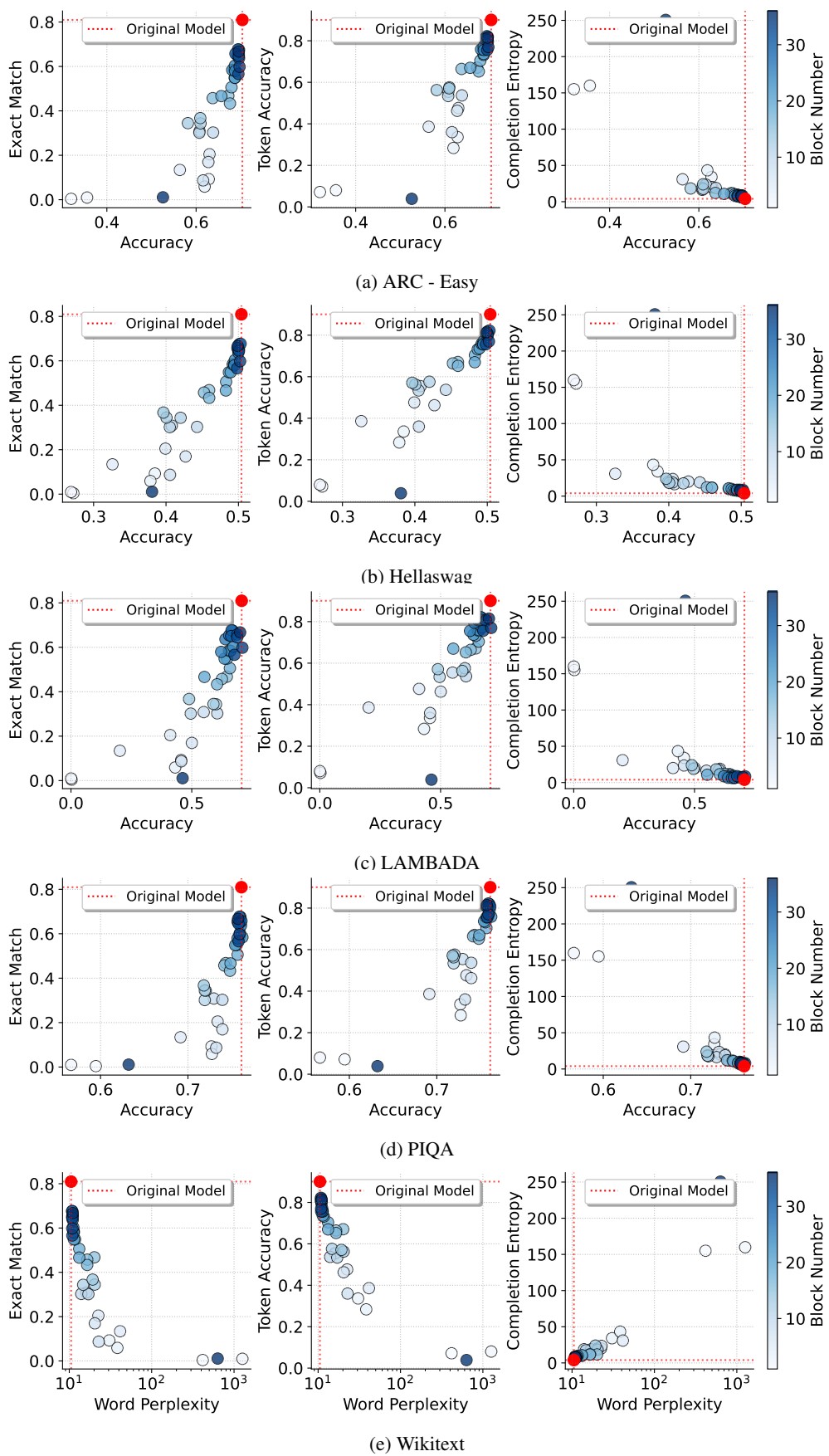

Figure A12: Memorization vs. Downstream Performance for Pythia-12B with short-circuiting applied to the attention mechanism of each block independently.

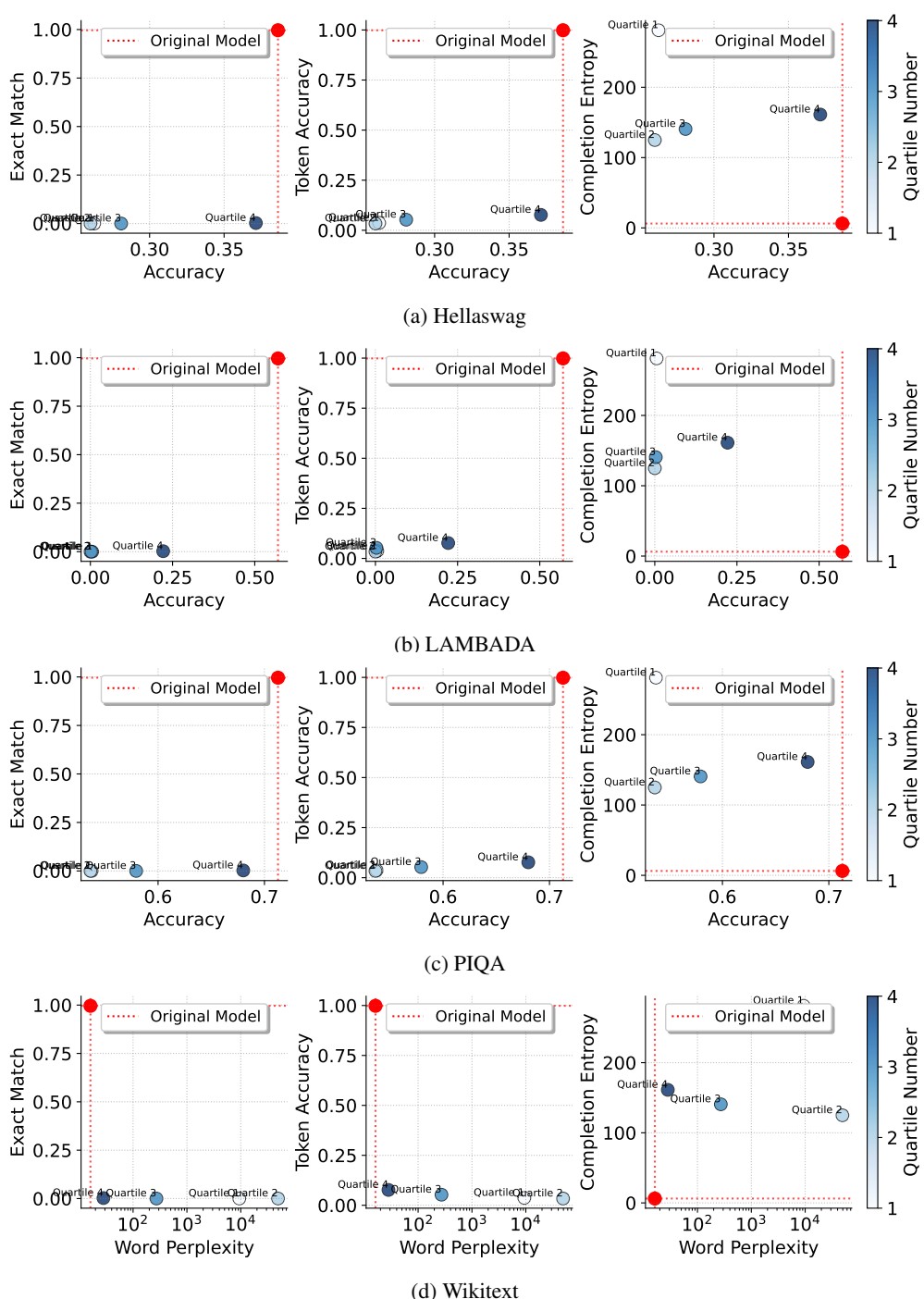

Figure A13: Memorization vs. Downstream Performance for GTPNeo-1.3B with short-circuiting applied to all attention blocks in each quartile of the model layers.

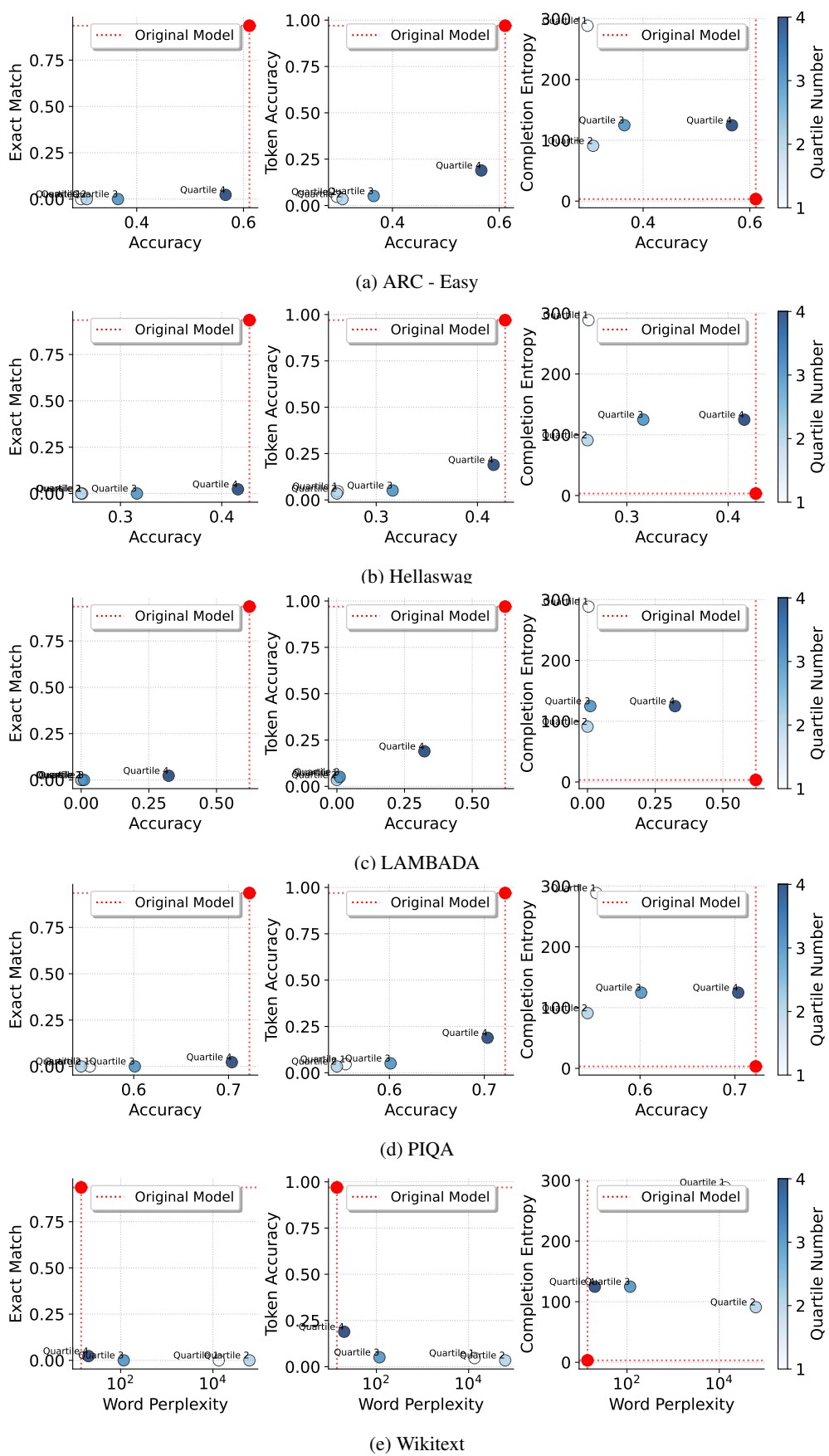

Figure A14: Memorization vs. Downstream Performance for GTPNeo-2.7B with short-circuiting applied to all attention blocks in each quartile of the model layers.

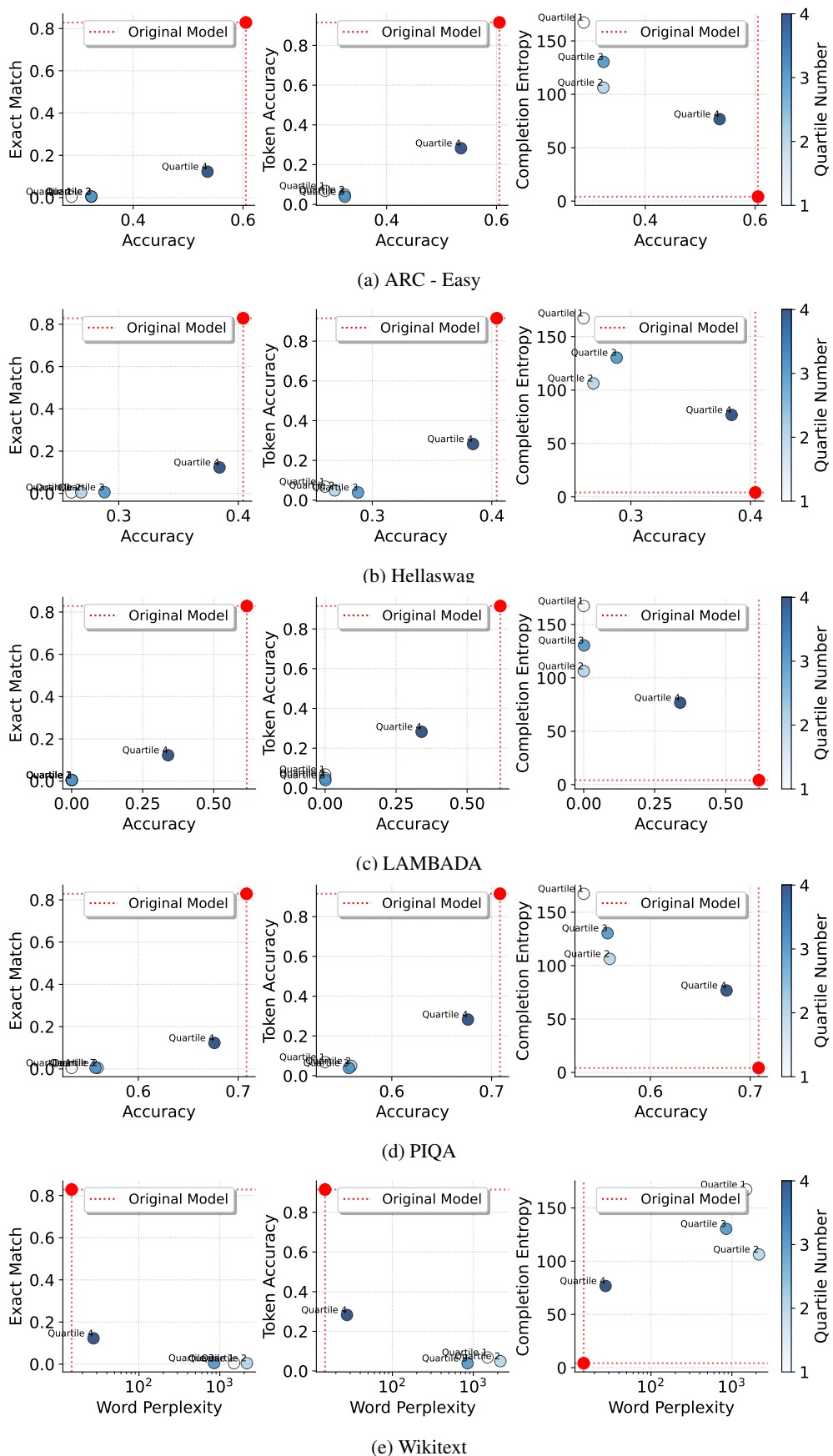

Figure A15: Memorization vs. Downstream Performance for Pythia-1.4B with short-circuiting applied to all attention blocks in each quartile of the model layers.

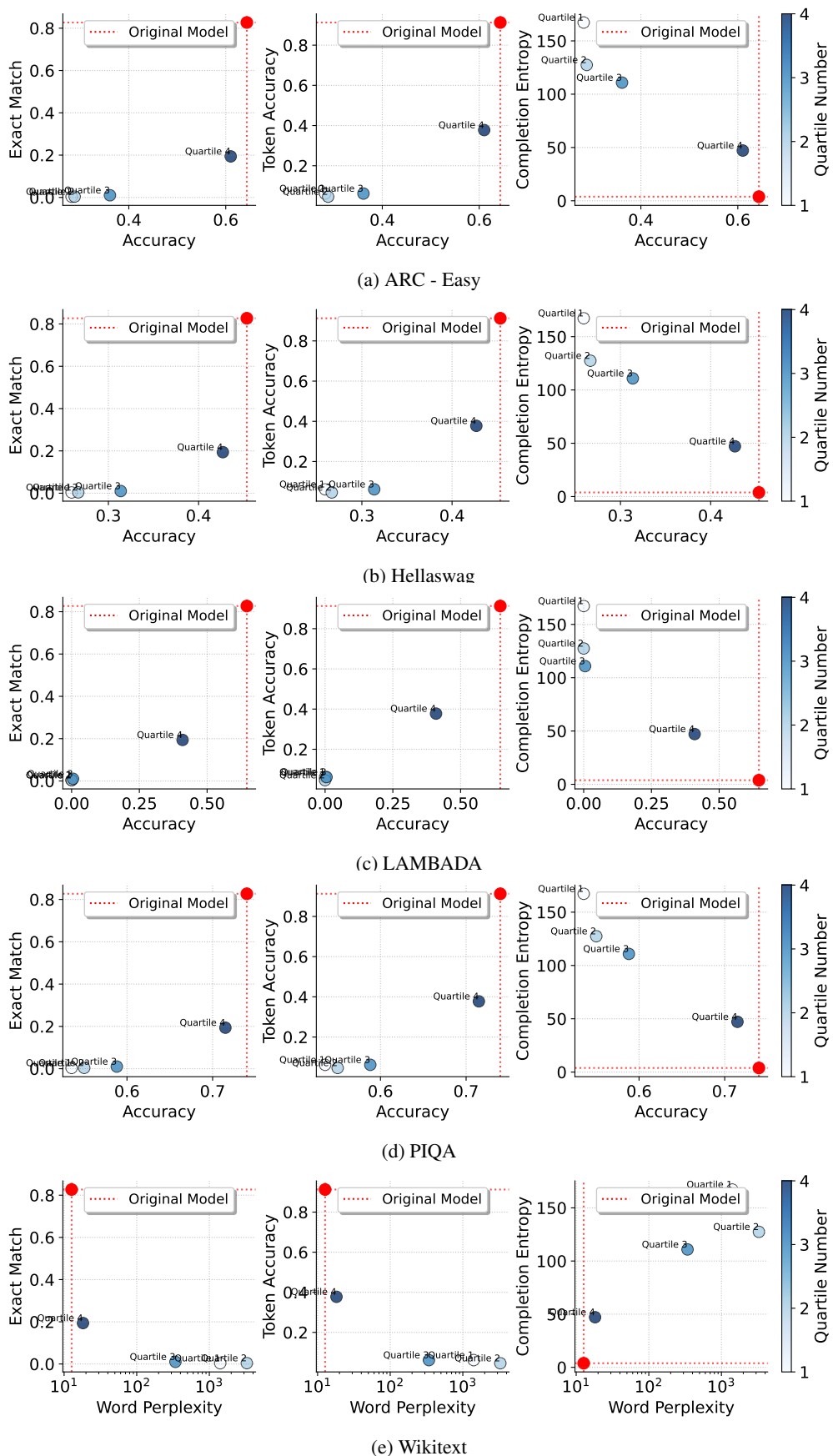

Figure A16: Memorization vs. Downstream Performance for Pythia-2.8B with short-circuiting applied to all attention blocks in each quartile of the model layers.

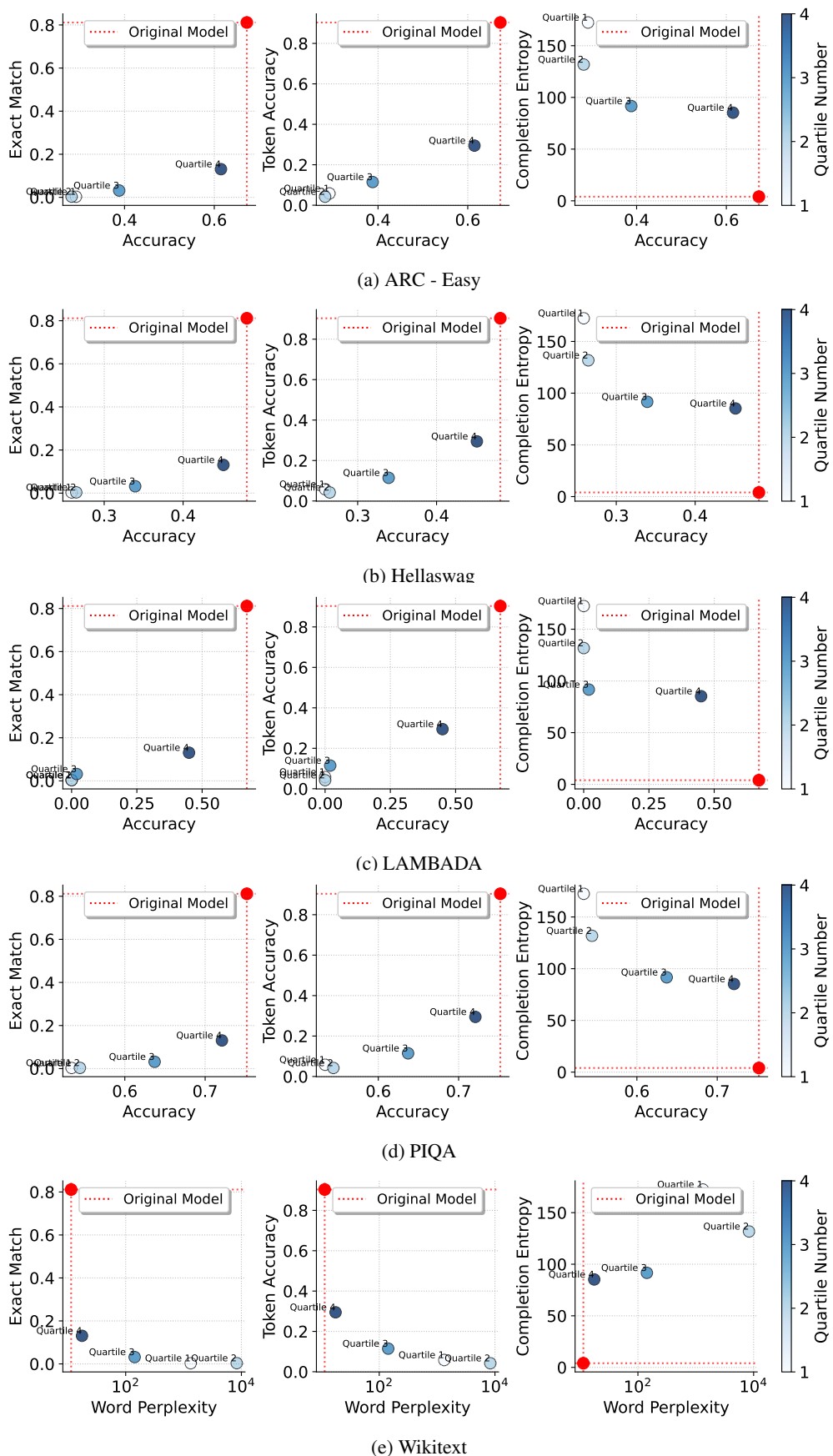

Figure A17: Memorization vs. Downstream Performance for Pythia-6.9B with short-circuiting applied to all attention blocks in each quartile of the model layers.

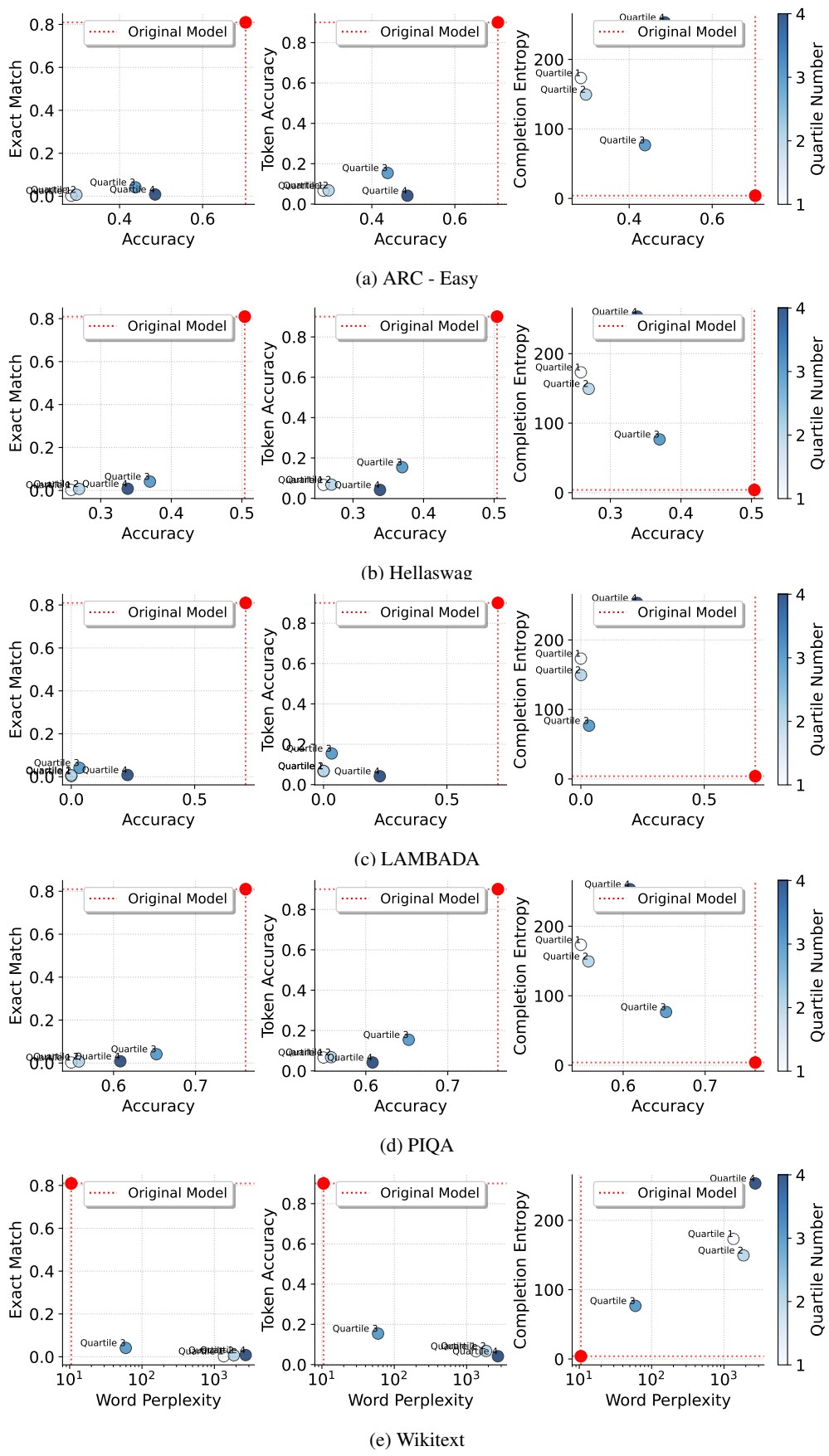

Figure A18: Memorization vs. Downstream Performance for Pythia-12B with short-circuiting applied to all attention blocks in each quartile of the model layers.

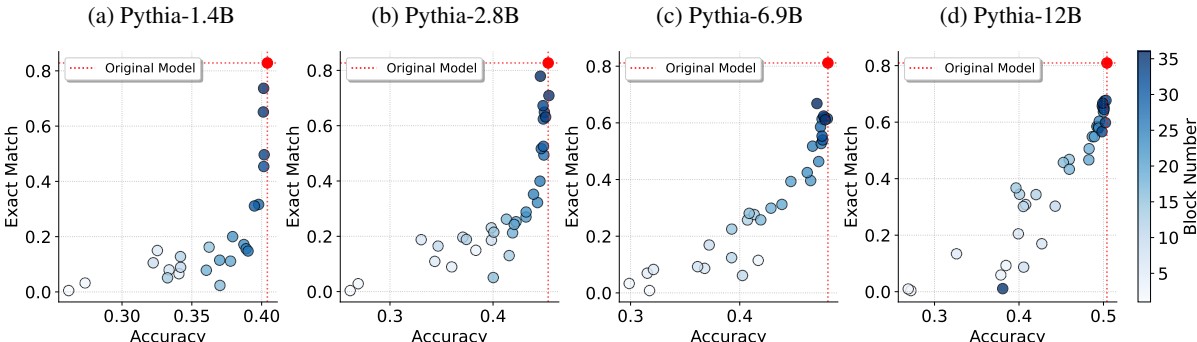

Figure A19: Results for short-circuiting the attention mechanism in each block of Pythia models across four scales - 1.4B, 2.8B, 6.9B, 12B for the HellaSwag benchmark

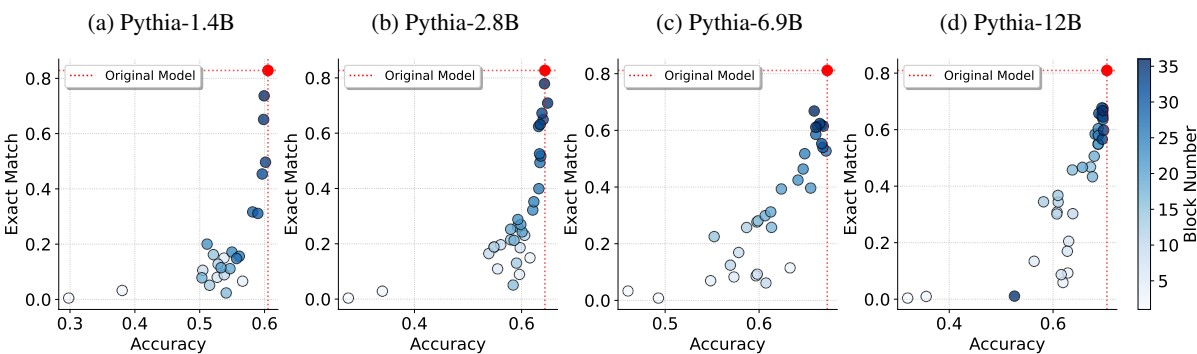

Figure A20: Results for short-circuiting the attention mechanism in each block of Pythia models across four scales - 1.4B, 2.8B, 6.9B, 12B for the ARC-Easy benchmark

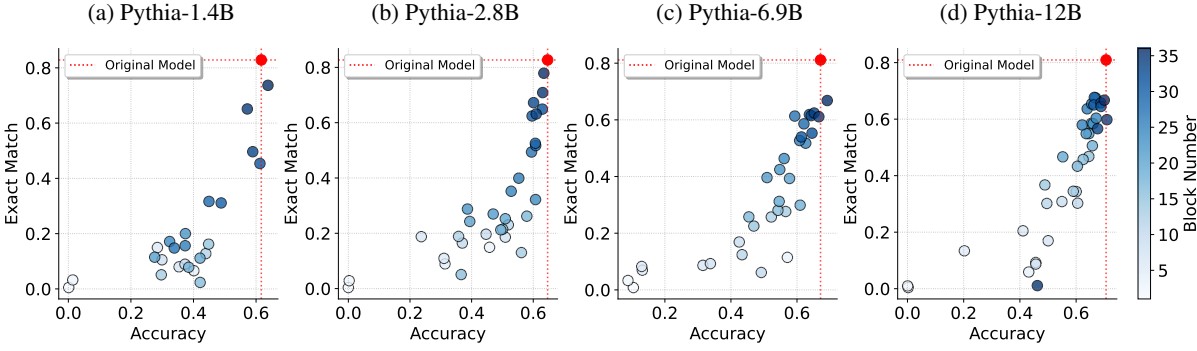

Figure A21: Results for short-circuiting the attention mechanism in each block of Pythia models across four scales - 1.4B, 2.8B, 6.9B, 12B for the LAMBADA benchmark

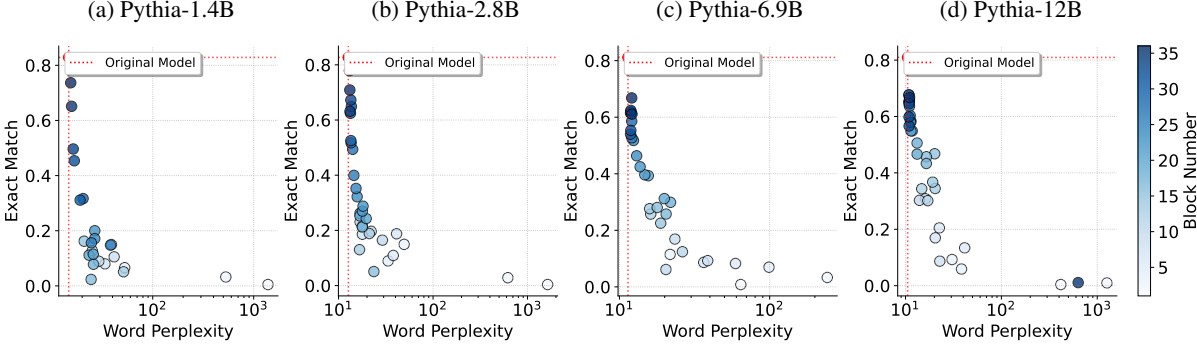

Figure A22: Results for short-circuiting the attention mechanism in each block of Pythia models across four scales - 1.4B, 2.8B, 6.9B, 12B for the Wikitext benchmark

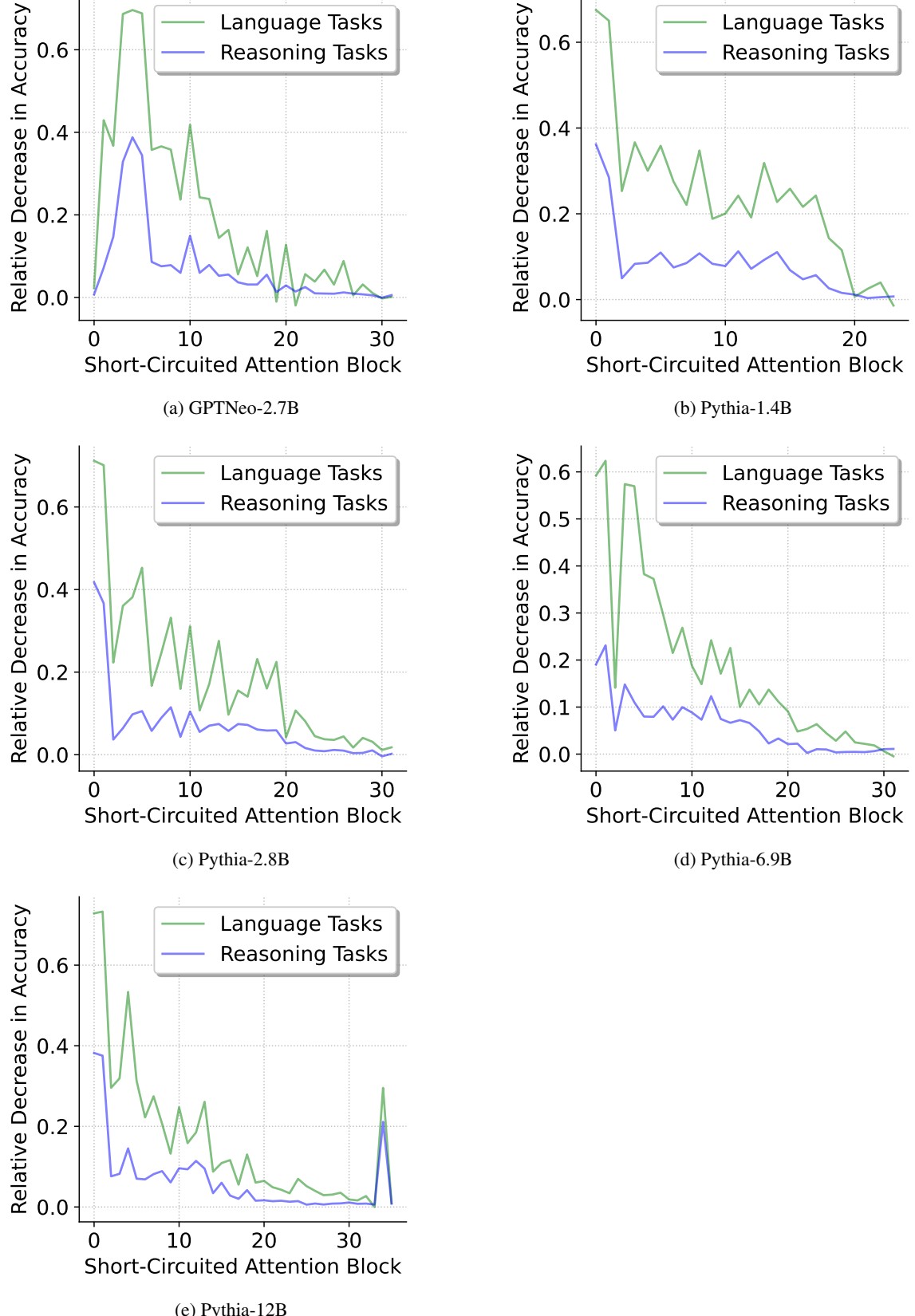

Figure A23: Relative decrease in benchmark performance for reasoning and language understanding tasks after short-circuiting attention mechanism in each block of (a) GPTNeo-2.7B, (b) Pythia-1.4B, (c) Pythia-2.8B, (d) Pythia-6.9B, and (e) Pythia-12B.