# OpenReview forum: "Analyzing Memorization in Large Language Models through the Lens of Model Attribution"
_ICLR.cc/2025/Workshop/BuildingTrust — BuildingTrust_

### Official Review · Reviewer_udMV · 2025-03-01
**Review: Analyzing Memorization in Large Language Models through the Lens of Model Attribution**

**Rating:** 7
**Confidence:** 4

**Review:**

The paper "Analyzing Memorization in Large Language Models Through the Lens of Model Attribution" explores the issue of memorization in LLMs and investigates architecture specific details of LLMs contributing to memorization. The authors focus solely on the attention mechanism, claiming reasoning of LLMs are largely derived from multiheaded self attention, as opposed to the FFN and other layers. The introduction is well written and the primer/ refresher on decoder only LLM architecture and the self attention mechanism is a nice touch, however a lot of terminology is not well defined in this section. The authors propose a novel "attention short circuiting" mechanism where attention layers at any given layer is replaced by an identity matrix multiplied by the value matrix, keeping the FFN and others untouched. Despite the narrow focus on the attention mechanism, analysis into architecture specific details contributing to memorization is a great study. The authors findings suggest that deeper attention layers contribute more to memorization, whereas earlier layers are essential for reasoning and generalization.

The approach is validated using Pythia and GPT-Neo model families on a handful of benchmark datasets, showing a practical method to reduce memorization while preserving performance. The results generalize across different scales of the Pythia model family, indicating that the technique is not only effective on smaller models but could also be feasible for larger models. While the experimental results are compelling, the focus on smaller LLMs, in particular LLMs which share the same huggingface attention layer (gpt-neo-x) is concerning as it does not represent a large enough sample to generalize to other LLMs. It is not convincing that the short circuit mechanism works in general, reducing memorization while keeping reasoning ability intact. It seems quite invasive to short circuit the attention layer, and while the results are compelling, t would be nice to see more extensive experimentation to support the claim.

The quality of this work is high, and the clarity is acceptable. The mathematical rigour is great and easy to read. While there are some limitations, the strengths outweigh them. The proposed approach has the potential to influence future research on ethical AI deployment and architectural understanding of memorization in LLMs.

---

### Official Review · Reviewer_FiZ3 · 2025-03-02
**Anonymous Review**

**Rating:** 7
**Confidence:** 4

**Review:**

The paper introduces a "short-circuit" method to bypass attention modules in transformers by replacing attention weights with that of the identity matrix. This is identical to using value (V) projection of the token as SDPA (Q, K, V) output. This intervention is used to bypass attn modules and study memorization in decoder block only transformers. Further, paper provides a theoretical framework explaining how disabling attention at different depths affects memorization and generalization. Additionally, the work also discusses an approach to mitigating memorization while maintaining generalization.

In their experiments, it is empirically shown that this short-circuiting of attention reduces memorization whilst impact on text generation performance is dependent on which layers' head is it applied on. Short-circuiting later blocks reduce memorization with minimal cost.

This paper aids the study of memorization and generalization trade-off and shows how attention module reacts to both with and without their method. Their method showcases acceptable performance in reducing memorization with minimal impact on generalization. The paper also matches the workshop and the venue.

---

### Official Review · Reviewer_f59i · 2025-03-03
**Focusing on Role of Attention Blocks in Memorization**

**Rating:** 6
**Confidence:** 4

**Review:**

This paper examines the role of attention blocks in implementing memorized sequences in large language models. They introduce a procedure of "short-circuiting" attention blocks and show that while initial attention blocks play a major role in implementing both general capabilities and memorization, later layers can be effectively "short circuited" (reverted to the identity) in order to mitigate memorization without harming general model capabilities. This is empirically tested via experiments on GPT-Neo and Pythia models. I think that the this paper has some interesting contributions: firstly most prior work focuses on examining the role of MLP components for storing memorization (in the form of key-value) pairs while this paper focuses on the attention mechanism. While the empirical findings are somewhat interesting, however, I find the theoretical analysis unconvincing. It seems to me that the primary conclusion of the theory is that the difference associated with short-circuiting an attention block accumulates across layers, which is unsurprising. While this explains the role played by later layer short-circuiting as a mechanism for roughly maintaining the original input distribution, it does not explain why memorization should necessarily compound in later attention blocks -- which is the primary claim made by this paper. I also believe that the paper could be stronger if the impact on general validation loss of short-circuiting attention blocks. This could provide another lens on the impact on general model performance. Finally, I think the paper would be more compelling if a stronger evaluation was used to measure memorization. Currently, the evaluation is simply exact match, however, this may provide a false sense of security against other prompting strategies that could elicit memorized sequences.

---

### Decision · Program_Chairs · 2025-03-04

**Decision:**

Accept

**Comment:**

Reviewers agree that the study is well-executed, novel, and relevant to understanding LLM memorization. Strengths include clear theoretical framing, experimentation, and practical implications for mitigating memorization while preserving generalization. However, some concerns remain about the generalizability of results to larger models and the theoretical explanations for why later layers contribute more to memorization.